# Temporally Rich Deep Learning Models for Magnetoencephalography

**Tim Chard**                                           *timothy.chard@hdr.mq.edu.au*
*School of Computing*
*Macquarie University*

**Mark Dras**                                              *mark.dras@mq.edu.au*
*School of Computing*
*Macquarie University*

**Paul Sowman**                                        *paul.sowman@mq.edu.au*
*School of Psychological Sciences*
*Macquarie University*

**Steve Cassidy**                                     *steve.cassidy@mq.edu.au*
*School of Computing*
*Macquarie University*

**Jia Wu**                                                     *jia.wu@mq.edu.au*
*School of Computing*
*Macquarie University*

**Reviewed on OpenReview:** *https://openreview.net/forum?id=zSeoG5dRHK*

## Abstract

Deep learning has been used in a wide range of applications, but it has only very recently been applied to Magnetoencephalography (MEG). MEG is a neurophysiological technique used to investigate a variety of cognitive processes such as language and learning, and is an emerging technology in the quest to identify neural correlates of cognitive impairments such as those occurring in dementia. Recent work has shown that it is possible to apply deep learning to MEG to categorise induced responses to stimuli across subjects. While novel in the application of deep learning, such work has generally used relatively simple neural network (NN) models compared to those being used in domains such as computer vision and natural language processing. In these other domains, there is a long history in developing complex NN models that combine spatial and temporal information. We propose more complex NN models that focus on modelling temporal relationships in the data, and apply them to the challenges of MEG data. We apply these models to an extended range of MEG-based tasks, and find that they substantially outperform existing work on a range of tasks, particularly but not exclusively temporally-oriented ones. We also show that an autoencoder-based preprocessing component that focuses on the temporal aspect of the data can improve the performance of existing models. Our source code is available at https://github.com/tim-chard/DeepLearningForMEG.

## 1 Introduction

Magnetoencephalography (MEG) is a brain imaging technique that detects magnetic fields generated by the brain at a high temporal resolution (Gross, 2019; Proudfoot et al., 2014). Like Electroencephalography (EEG), it has applications in diagnosing neuropsychiatric disorders; one key clinical application is epileptic source localisation (Stufflebeam, 2011). MEG, unlike EEG which captures the neuronal activity by recording

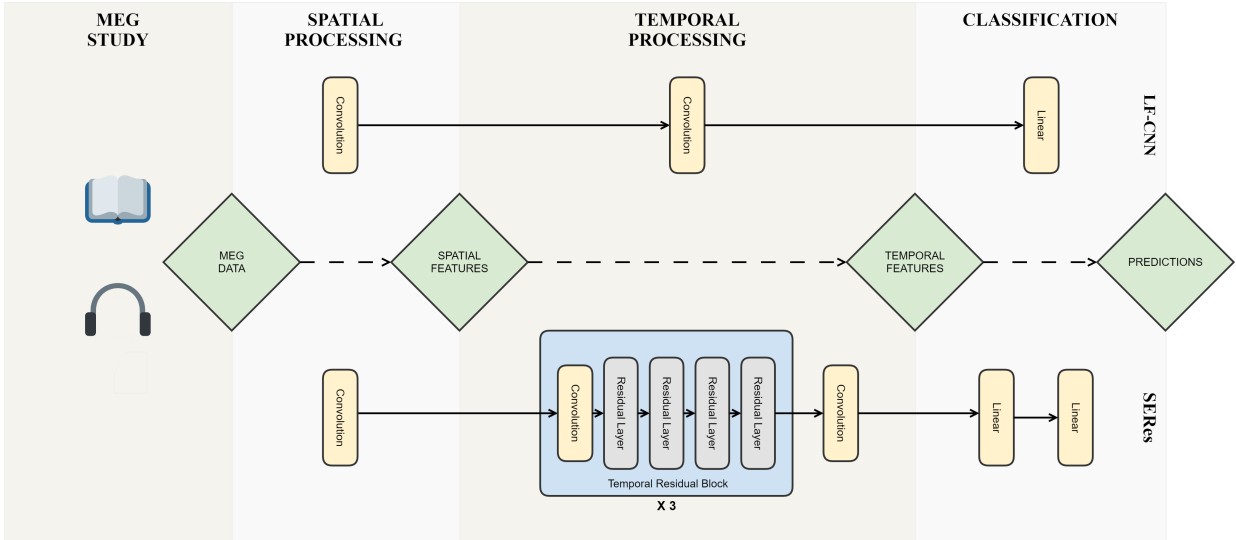

Figure 1: The four components of our new model architecture for handling temporally rich data: the MEG study data, the spatial and temporal processing of the data, and finally the classification. Our SERes architecture is focused on extracting temporal relationships within the data; compared to the LF-CNN of Zubarev et al. (2019), for example, it is capable of extracting much more complex relationships in the data.

the electrical potential on the surface of the scalp, detects the magnetic flux that is associated with electrical brain activity (Hari & Puce, 2017). As magnetic fields pass through the skull relatively unaltered, MEG has the relative advantage of capturing neuronal activity with less spatial mixing of the underlying signal (Proudfoot et al., 2014; Brookes et al., 2011; Luckhoo et al., 2012).

The magnetic flux that is captured by MEG is in the picoTesla and femtoTesla range, which is more than 7 orders of magnitude smaller than Earth's ambient magnetic field (Hari & Puce, 2017). Notwithstanding precautions to shield these machines from all other sources of magnetic fields, MEG still has a low signal-to-noise ratio. It is thus not surprising that it has been a challenge to process such data, and in particular to apply machine learning to it.

Early work on classifying MEG signals (Georgopoulos et al., 2005; Waldert et al., 2008) used relatively simple linear models. Conventional machine learning techniques like Support Vector Machines were later employed by, for example, Quandt et al. (2012) (with mixed success) and Westner et al. (2018). It was only in 2019 that deep learning was first seriously applied to MEG analysis (Zubarev et al., 2019; Kostas et al., 2019; Aoe et al., 2019), showing the superiority of end-to-end deep learning over earlier feature engineering, with some further work since (Dash et al., 2020; Pantazis & Adler, 2021; Huang et al., 2021; Giovannetti et al., 2021; Lopez-Martin et al., 2020; Ovchinnikova et al., 2021). In general these applied ideas from image processing and computer vision, and specifically the Convolutional Neural Network (CNN) architecture, similarly to the earlier EEGNet (Lawhern et al., 2018).

Some of this prior work focuses on analysis of resting state data, typically for predicting neurological issues (Aoe et al., 2019; Huang et al., 2021; Giovannetti et al., 2021); other work such as Pantazis & Adler (2021) focuses on improvement of tools, source localisation or various kinds of signal cleaning. The focus of the present work is what Roy et al. (2019), in their survey of deep learning in EEG, characterise as "active Brain-Computer Interface (BCI) classification", covering activity such as motor imagery, speech decoding and mental tasks. The three core works in this category for MEG are Dash et al. (2020), which aims to classify real and imagined speech utterances; Kostas et al. (2019), which although it aims to predict the static property of subject age, does so on the basis of mental tasks and speech utterances; and Zubarev et al. (2019), which examines a number of motor imagery and mental tasks. Among all of these, where more complex systems from computer vision are used directly, it is not generally in an end-to-end architecture:

Giovannetti et al. (2021) used AlexNet, but only for feature extraction rather than end-to-end learning, while Dash et al. (2020) used AlexNet and two others pretrained on image data, where the MEG data had to be preprocessed into scalograms to fit those systems.

Otherwise, the models involve relatively simple architectures, such as a spatial de-mixing layer followed by a temporal convolution layer (Kostas et al., 2019; Zubarev et al., 2019). The spatial de-mixing layer applies a set of spatial filters to the raw input, which separates spatial patterns into higher-level features. The next layer then identifies rudimentary temporal patterns in these spatial features. Despite the relative simplicity, these works all found that their models performed better on average than baseline conventional machine learning models like Support Vector Machines. Zubarev et al. (2019) also compared against domain-specific and general computer vision neural networks trained end-to-end, and found that their models performed better than those, which suggested that the much more complex architectures from computer vision and elsewhere do not necessarily suit this domain.

However, only one of the datasets used by Zubarev et al. (2019), Cam-CAN (Shafto et al., 2014), is a large one, and it is well-known that deep learning approaches need much larger amounts of high-quality data than conventional machine learning; it was on this dataset that their computer vision architecture, VGGNet (Simonyan & Zisserman, 2015), performed competitively. Moreover, the classification task on the Cam-CAN dataset was an easy one that does not require sophisticated models to perform well. Indeed, all approaches tested achieved well over 90% accuracy, and none performed statistically significantly better than any other, thus not allowing a true determination to be made about the superiority of different models. In this paper we extend the application of deep learning to MEG prediction tasks by considering both richer architectures suited to the data, and additional dataset tasks that are suited to clear evaluation of these architectures. In particular, as we observe below, these richer architectures need to better incorporate the temporal aspects of the data. This involves three challenges.

**Challenge #1: Lack of Focus on Temporal Aspects of Data** The models from existing work, with relatively few convolutional layers (two in the case of LF-CNN and VAR-CNN of Zubarev et al. (2019)), are limited in the type of interactions that can be expressed; compare this with the 19 convolutional layers of the VGG-19 model used as a comparison by Zubarev et al. (2019), or the many more layers in later computer vision architectures. Further, they focus most heavily on spatial relationships in the data, and it is not uncommon for a model's temporal layers to learn spatio-temporal relationships, as is the case in VAR-CNN of Zubarev et al. (2019) and SCNN and Ra-SCNN of Kostas et al. (2019). However, with MEG the signal can only be recorded at a discrete number of locations around the head, so that the generated dataset is spatially sparse. It also partially violates an implicit assumption in computer vision architectures, that adjacent values in the data are a result of adjacent values in the source: in MEG data this assumption is only strictly valid in individual channels. This means that it is difficult to develop a model that is able to learn a hierarchy of spatial features of the kind that are learnt by state-of-the-art computer vision architectures. In contrast, the data is dense along the temporal dimension (often sampled at around 1KHz). We therefore propose architectures that focus on building up a hierarchy of temporal features that are combined using residual connections, a popular method that was first used in computer vision (He et al., 2016).

The first part of this idea, of a hierarchy of temporal features, is drawn from WaveNet (Oord et al., 2016), a speech recognition and synthesis model that is capable of tracking very long temporal dependencies. These long-term dependencies are common in speech: for instance, in the sentence "he was sitting down because he hurt his leg", the gendered pronouns can have an arbitrary distance between them. However, temporal patterns are something that existing models have not really exploited at all. Our models are therefore designed to be capable of learning more complex temporal relationships.

The second part of the idea, of residual connections, has become a standard in image processing. While much of the computer vision research is focused on learning spatial relationships more effectively, there has also been a significant amount of more general work that can be applied to many domains. One such development was in the ResNet architecture, which introduced the residual connection (He et al., 2016). The residual connections are layers that learn the best way to *alter* the input to reduce superfluous data, and these layers have allowed much larger neural networks to be trained as a result. Fig 1 gives schematic representation of our architecture.

**Challenge #2: Use of Raw Input Data in End-to-End Models** While it has conventionally been common in brain signal analysis to use different forms of dimensionality reduction (including now using deep learning), the above end-to-end models, existing and proposed, take the raw data as input to the neural network. In other domains, however, the generation of intermediate latent representations via deep learning has been found to be more useful, particularly where these can be learnt from large amounts of unlabelled data. (For instance, in Natural Language Processing (NLP), the use of contextual language models trained on unlabelled data as a starting point for downstream tasks like classification or machine translation has been shown to greatly improve performance on those tasks (Devlin et al., 2018; Peters et al., 2018).)

One possible approach to learning these intermediate latent representations is via autoencoders (Masci et al., 2011). In encoding an input into a lower-dimensional latent representation and then attempting to reconstruct the original input from this reduced representation, autoencoders learn to encode only relevant information in these representations which can then be used as input for downstream tasks.

In this paper, we propose an autoencoder that is only capable of learning temporal features, for use as a preprocessing component for simpler models, as an alternative to the entirely new architectures outlined above. In addition to separating out the processing of temporal information in a way that would allow it to be used with future spatially-oriented architectures, it also allows us to examine our assumption about the value of temporal features in MEG processing.

**Challenge #3: Lack of Activity Detection Tasks on Large Datasets** As noted, deep learning requires very large amounts of high-quality data; in the past, the creation of datasets such as ImageNet (Deng et al., 2009) have been pivotal for their domains. The Cam-CAN dataset, as currently released, only includes data for a single task where subjects were exposed to either a visual or auditory stimulus: given the difference in brain regions responsible for each of these, the simplicity of the task explains the high accuracy of all models in Zubarev et al. (2019) on that dataset. Applying different architectures to a more difficult task would allow scope for differentiating their performance.

Recently, another large dataset has been released, the Mother Of Unification Studies (MOUS) (Schoffelen et al., 2019), in which subjects are presented a series of words either visually or via audio. The words could form a sentence which is easy to process because of its syntactic structure, a sentence which is hard to process, or an arbitrary list of words. This leads to a variety of classification tasks with a range of difficulty. For instance, can we distinguish audio from visual stimuli? Can we distinguish stimuli corresponding to syntactically valid sentences versus a random ordering of those words? Can we distinguish syntactically more complex sentences from syntactically simpler ones? We can perform the same task that is used on Cam-CAN (i.e. distinguishing audio from visual stimulus), but the other classification tasks should prove to be much more challenging.

**Contributions** This paper makes the following contributions.

- We present several new deep learning architectures designed to suit the temporally rich, spatially sparse nature of MEG data (addressing Challenge #1).

- We present an autoencoder for use with previous models in preprocessing the temporal aspect of MEG data, and show that these can improve the performance of previous models on relevant tasks (addressing Challenge #2).

- We compare these experimentally with previous work, both on prior data and on a new large dataset with a range of more challenging classification tasks (addressing Challenge #3).

- We show that these new architectures clearly outperform the previous state of the art on tasks where temporal information is important; and we analyse which of the models best suits which kind of classification task (also addressing Challenge #3).

## 2 Background

Neurons in the brain rely on electrical currents for their functioning; by recording this electrical activity we will be capturing aspects of brain activity Two related technologies are able to capture these electrical

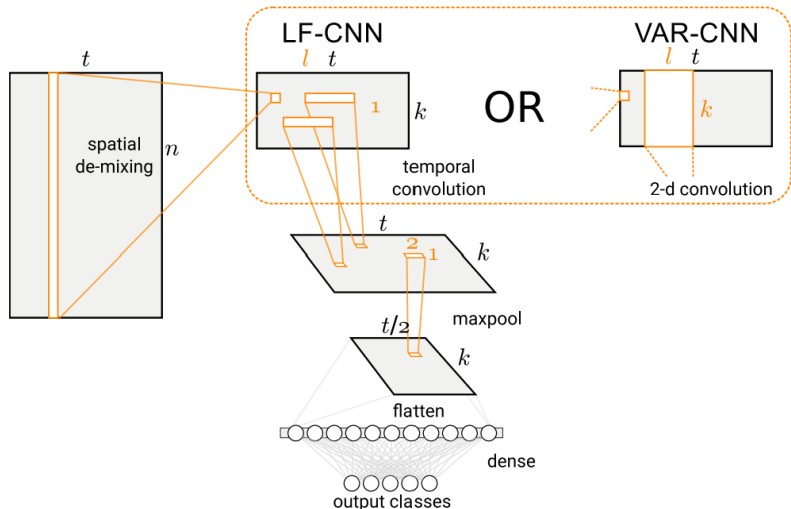

Figure 2: LF-CNN and VAR-CNN architectures from Zubarev et al. (2019).

impulses: Magnetoencephalography (MEG) and Electroencephalography (EEG). EEG captures the activity by recording the electrical potential on the surface of the scalp, while MEG detects the magnetic flux that is created by electric fields (Hari & Puce, 2017). Both MEG and EEG have a high temporal resolution (often sampled at around 1KHz) but are not known for their spatial resolution. This contrasts with functional magnetic resonance imaging (fMRI) which provides high spatial resolution but can not be reliably used to study rapid changes in the brain.

Early work on predicting human activity from MEG data were motivated by Brain-Computer Interfaces (BCIs) that could operate without invasive brain-monitoring technology. Georgopoulos et al. (2005) and Waldert et al. (2008), for example, looked a predicting hand movement trajectory; the former looked only at MEG signals and used linear summations of weighted MEG components for prediction, while the latter considered both MEG and EEG signals and produced predictions from regularised linear discriminant analysis. Subsequent work extended this to conventional machine learning techniques. Quandt et al. (2012), for example, looked at predicting finger movement, also considering both MEG and EEG signals, using them as input to a linear Support Vector Machine (SVM). They found that the use of MEG signals led to better performance and attributed the result to the better spatial resolution. Other work like Westner et al. (2018) looked at a different stimulus: their task was to distinguish auditory from visual presentation of words, using Random Forests for the prediction.

While related, the literature on using EEG is more extensive, probably because of the relatively greater accessibility of the technology: a comprehensive review of work applying machine learning to EEG data can be found in Lotte et al. (2018). EEG has consequently seen earlier applications of deep learning: Roy et al. (2019) gives a survey of this, and a standard open source tool based on Convolutional Neural Networks, EEGNet (Lawhern et al., 2018), is available. However, it is only more recently that deep learning has been applied to MEG.

One strand of work involves only resting state data, typically for clinical applications, such as Aoe et al. (2019) and Giovannetti et al. (2021), both of which aim to identify neurological issues in subjects. Aoe et al. (2019) proposed MNet, a CNN consisting of 10 convolutional layers to distinguish patients with epilepsy or spinal injury from healthy patients; Giovannetti et al. (2021), aiming to detect dementia, proposed Deep-MEG, which used AlexNet (Krizhevsky et al., 2012) to extract functional connectivity (FC) indices as features in an ensemble approach.

Of the work directly relevant to the focus of the present paper — tasks that would fall under the active BCI classification of Roy et al. (2019) — Dash et al. (2020) used data from 8 subjects, where those subjects were

presented with five possible phrases that they were required to utter; the aim was then to predict which is the chosen phrase at several possible stages (pre-stimulus, perception, preparation and production). They used two kinds of models for this prediction, a single fully-connected neural network with one hidden layer, and three much richer architectures from computer vision (AlexNet (Krizhevsky et al., 2012), ResNet101 (Wu & He, 2020), and Inception-Resnet-v2 (Szegedy et al., 2017)). These latter architectures were pretrained on ImageNet (which contains images of everyday objects like cars), and so to be compatible with this type of input the MEG signal was converted to scalograms based on (spatial-)spectral-temporal features, and these were converted to the necessary image dimensions for the architectures. Another work, Kostas et al. (2019), aimed to detect the age of subjects performing a speech production task, with the primary dataset for the simplest task (eliciting the phoneme */pah/*) consisting of 89 subjects, and the most challenging (eliciting a verb) consisting of a subset of those of size 28. For the age prediction, they developed two deep neural network models: the SCNN, a CNN which consists of a spatial filtering component with several convolutions, followed by a temporal component also consisting of several convolutions; and the Ra-SCNN, which adds an LSTM and attention mechanism to the end of the temporal component. These models, unlike Dash et al. (2020) but like our own work, take the raw MEG signal as input. The closest to our own work is that of Zubarev et al. (2019), who proposed two deep learning architectures, LF-CNN and VAR-CNN (Fig 2), which were based on the generative model of non-invasive electromagnetic measurements of the brain activity of Daunizeau & Friston (2007). They also developed methods for analyzing the models' predictions, where they looked at the spatial and temporal features that were most strongly related to each predicted class. This allowed them to visualise how these networks derived their prediction and how these prediction were associated with regions of the brain.

They performed four different experiments that looked at three different tasks, and focused on generalization to new subjects. The experiments covered a five-class classification task, distinguishing five different sensory stimuli (visual, auditory and electrical) on a private dataset of 7 subjects; a three-class motor imagery task on a private dataset of 17 subjects, on both the static dataset and a version with real-time data; and a two-class task distinguishing a visual stimulus from an audio stimulus, using the large-scale public Cam-CAN dataset (Shafto et al., 2014), from which they used 250 subjects. These experiments were in the context of BCIs, and so in addition to the standard validation and test accuracy for evaluation they also simulated a real-time brain-computer interface environment. This was similar to test accuracy except the model parameters were updated after it had made a prediction for each trial and aimed to gauge how well the models could adapt to new subjects as the subject is using the BCI. Comparison systems were two SVMs (linear and RBF kernels), two deep learning systems for EEG analysis (EEGNet (Lawhern et al., 2018) and ShallowFBCSP-CNN (Schirrmeister et al., 2017)), and a computer vision model (VGG19 (Simonyan & Zisserman, 2015)); the comparison deep learning systems generally have more complex architectures than LF-CNN and VAR-CNN.

On the first three experiments, with small datasets, the comparison deep learning models all performed poorly: for instance, in Experiment 1, the comparison deep learning models scored between 60.1% and 76.8%, against 80.2% and 82.7% for the SVMs and 83.1% and 85.9% for LF-CNN and VAR-CNN respectively. In contrast, on the large Cam-CAN dataset, the lowest accuracy was 92.1% for the linear SVM, with EEGNet and VGG19 at 93.0% and 92.3% respectively, and LF-CNN and VAR-CNN 95.1% and 95.8% respectively. This is in line with the general understanding that deep learning methods require more data for reasonable performance (see e.g. Brigato & Iocchi (2021) for discussion). This suggests that with large datasets more complex architectures are a promising line of research.

## 3 Models

In this section, we propose a number of different network architectures that are capable of modelling more complex relationships than the models discussed in §2. Our first architecture, TimeConv, is the simplest one that explores the importance of temporal relationships in the data. These ideas are then refined into a spatially invariant autoencoder architecture (TimeAutoencoder) that is built with a number of temporal residual blocks which exploit these temporal relationships with residual connections (He et al., 2016). Our third model (SERes) combines the same temporal residual blocks with the spatial relationships that have already been found to be effective in the models of Zubarev et al. (2019). Our final new architecture (SETra) uses the Transformer architecture (Vaswani et al., 2017) which follows from the way that we have thought

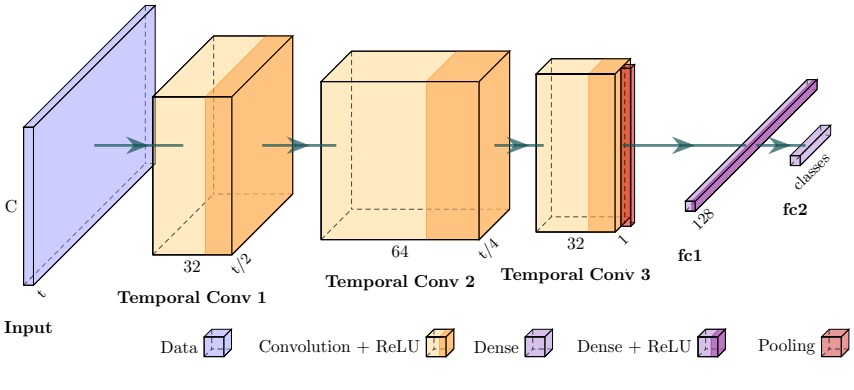

Figure 3: The TimeConv architecture.

about the MEG image as a sequence of spatial embeddings. Following these, we propose a preprocessing component based on an autoencoder (TimeAutoencoder) that can be combined with LF-CNN and VAR-CNN to improve their ability to use temporal information.

## 3.1 TimeConv

The input to these models consists of a 2-dimensional matrix, one representing time and the other representing the spatial aspect of the data. We can think of this data as a monochrome image, where spatial dimension (the MEG channels) corresponds to the height and the temporal dimension is the width. This architecture is designed to help gauge the importance of temporal relationships in the data. As such, a separation between temporal and spatial relationships is enforced throughout the body of the network, such that the network learns spatial features only at the final fully-connected layers. Fig 3 shows the high-level structure of the network.

Even though the main body is composed of just three temporal convolution layers, this is already deeper than most previous work. Each temporal convolution layer consists of a 2D convolution (a convolution that is applied over the height and width of the input), limited to act only on a single meg channel; this encourages the network to focus on temporal aspects of the data and effectively disallows it from learning spatial features. These convolutions have a kernel size of $(1, 5)$, dilation of $(1, 3)$ and a stride of $(1, 3)$ and use "same" padding, and are followed by a standard ReLU activation function. There are 32 filters in the first layer, 64 in the second, and 32 in the last. These settings were chosen to allow for more expressiveness than the models of Zubarev et al. (2019) while also keeping the number of elements in the output of the last layer relatively small. These layers are followed by a max-pooling layer, which means that ultimately the network is learning a non-linear function that maps the activity in each channel into a 32 dimension temporal embedding (the number of filters in the last layer). These embeddings are then concatenated and fed into a dense layer with 128 output features, a ReLU and finally an output layer with the number of classes as its dimension.

## 3.2 Temporal Residual Block

The Temporal Residual Block is a self-contained logical group of layers; we will use these in different configurations of these blocks in SERes and the TimeAutoencoder below. The structure of this block follows from the ideas in TimeConv, but unlike that model, the Temporal Residual Block is completely prevented from learning any spatial features. This means that this block is completely independent of the size of the spatial embeddings (or the number of MEG channels) in the input.

We implement this by with a 2D convolution that uses kernel of size $(1, 3)$. This has two consequences. First, because the kernel is effectively 1-dimensional, it is incapable of learning any spatial relationships. Second,

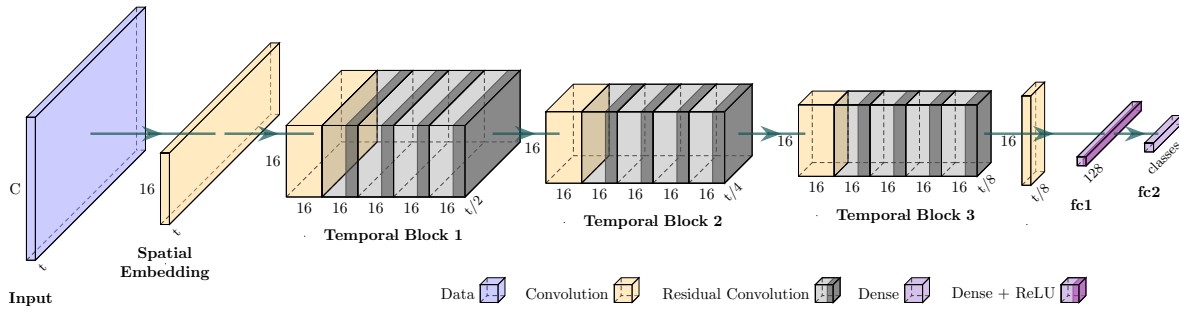

Figure 4: The Spatial Embedding Residual (SERes) network architecture.

because it is a 2D convolution, the kernel is applied to all channels, which encourages the network to learn low-level relationships that exist in all channels.

This convolution has some similarities to the LF-CNN, in that they both use a 1-dimensional kernel, but they are significantly different in the type of patterns they are able to learn. Consider how these convolutions will act on a black and white image, assuming the same number of kernels and same kernel size (i.e. both will have the same number of trainable parameters). Both architectures will apply a 1-dimensional kernel to each row of the image. However, the LF-CNN will apply a different kernel to each row, whereas our convolution will apply each kernel to all rows.

Each block consists of four Temporal Residual Convolution layers which are similar to residual layers used by He et al. (2016). However, we reduce the kernel to 1-dimension which means each convolution uses a kernel size of $(1, 3)$, stride of $(1, 2)$ and padding of $(0, 1)$. We use the same number of filters in each layer which we leave as a hyper-parameter. Our implementation of the residual connection is very similar to the PyTorch implementation (Paszke et al., 2019) and, ignoring the convolution that is being used, only differs in how batch normalization is applied. While we use a single batch normalization layer before either convolution, the PyTorch implementation uses two layers directly after each convolution.

### 3.3 SERes

While TimeConv learnt temporal relationships in the raw data, the temporal components of both the LF-CNN and VAR-CNN have operated on spatial embeddings which are able to capture when certain regions of the brain are active at the same time. The SERes architecture (Fig 4) combines both of these ideas, by first learning a spatial embedding which effectively reduces the number of "channels" to 16. So given an input $(1, C, t)$ with $C$ MEG channels and $t$ time samples, the spatial embedding layer will produce an embedding of size $(1, 16, t)$. This spatial embedding is then processed by three temporal residual blocks. Each of the three blocks has 16 filters and reduces the number of time samples by a factor of eight, leading to an output of $(16, 16, t/8)$. To reduce the number of parameters and avoid problems like overfitting, we apply a convolution with a single $(1 \times 1)$ kernel and outputs $(1, 16, t/8)$ and is then flattened to a $16 \times t/8$ feature vector. The head of the network is very simple and consists of two layers. The first has 128 output features and uses the ReLU activation. The last predicts the weighting of the classes which are used as part of the cross-entropy loss, which combines the softmax activation and the negative log-likelihood loss.

### 3.4 SETra

The evolution and impact of Transformer architectures (Vaswani et al., 2017) in machine learning have been substantial, with their application now extending beyond the initial natural language processing (NLP) domain into areas such as computer vision.

A key strength of Transformers lies in their ability to process information from an entire input sequence, unlike conventional convolutional networks that are limited to exploiting data within a specific receptive field of neurons. This limited scope means that a convolutional network can only integrate details from distant parts of the input as the receptive field progressively expands through deeper layers.

However, applying attention to each pixel in an image in a naive manner becomes computationally infeasible as the image size increases because the computational cost grows quadratically with the number of pixels. This has led to the development of innovative methods and architectures to manage this computational complexity.

Recent years have witnessed the emergence of architectures such as Vision Transformer (ViT) (Dosovitskiy et al., 2021), Image Generative Pre-trained Transformer (I-GPT) (Chen et al., 2020), Swin Transformer (Liu et al., 2021), and Cross-Covariance Image Transformer (XCiT) (Ali et al., 2021) that have successfully incorporated Transformers in computer vision tasks. These models have introduced various strategies like patch-based attention, hierarchical structures, or sophisticated mechanisms to make attention more computationally efficient. A detailed exploration of these models can be found in (Khan et al., 2022) as well as other approaches to optimize the computational cost of attention itself such as in Longformer and Reformer (Kitaev et al., 2020; Beltagy et al., 2020).

In our work, we address the computational challenge with a different approach. Our problem formulation as a sequence of spatial embeddings inherently results in sequence length scaling linearly with image width, unlike the quadratic scaling seen in other methods. To illustrate, for an input of size $(224 \times 224)$, while the ViT would split this into $14^2$ patches of $(16 \times 16)$ for attention, our method would attend to all 224 columns (or rows). This circumvents the need for any form of downsampling, while still remaining computationally feasible.

Distinguishing features of our SETra architecture compared to other Transformers include its size and training methodology. Despite their impressive flexibility, Transformers' ability to learn intricate relationships can lead to overfitting. To counteract this, our models are significantly smaller than even the smallest models used in other contexts. Moreover, while most Transformers are pre-trained using a next token prediction task, our SETra is trained directly on the classification task, consistent with other models evaluated in this work.

Similar to the previous model, we employ a spatial embedding of size 16, an equivalent to a "word embedding" in the NLP context. Our model includes four transformer layers as implemented by PyTorch (Paszke et al., 2019), each with an embedding size of 16 and a feedforward dimension of 64. Despite these dimensions being much smaller than typical, they are necessary for the effective training of our model.

### 3.5 TimeAutoencoder

As noted above, the TimeAutoencoder differs from the above new architectures in that it is a preprocessing component to be used by LF-CNN and VAR-CNN. It has two main components, the encoder and the decoder. Fig 5 shows these components separated by the latent representation that captures temporal information. Given an input with size $(1, C, t)$ where $C$ is the number of channels and $t$ is the number of time steps, this will be compressed down to $(1, C, t/2)$ before the original input is reproduced.

The first layer in the encoder increases the number of filters to $(32, C, t)$, which is then passed to Temporal Residual Block with 32 filter layers, maintaining the same dimensions. The last layer in the encoder is a strided temporal convolution that forms the bottleneck in the network. This layer uses the same settings as the other layers in the Temporal Residual Block, but in addition uses a stride of $(1, 2)$. Due to the stride, it outputs the latent representation which has half the number of time steps of the input but is otherwise the same $(1, C, t/2)$.

The decoder is similar to the encoder but replaces the first and last layer convolutions with transpose-convolution equivalents. The first layer will increase the number of filters to 32 so that the data has a shape of $(32, C, t/2)$ and like the encoder, this is input to the Temporal Residual Block. The last layer in the decoder is responsible for recreating the missing time steps and reproducing the input.

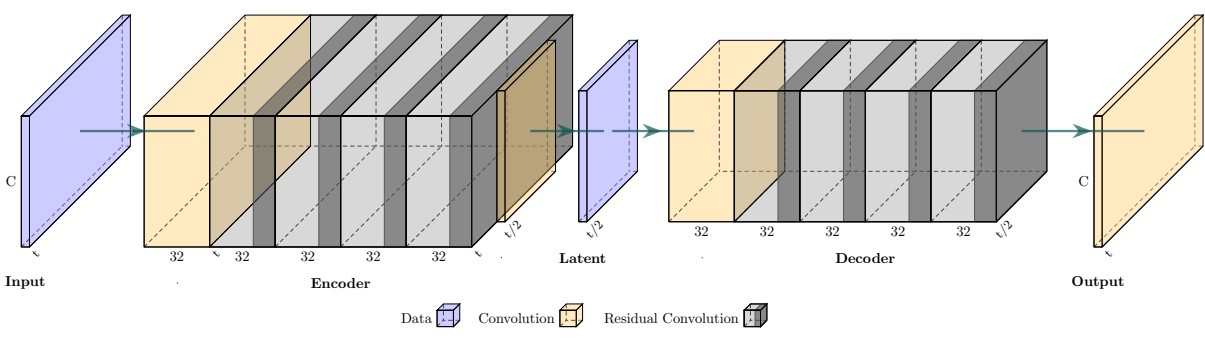

Figure 5: The TimeAutoencoder architecture.

## 4 Experimental Setup

In this section, we describe the datasets and the classification tasks derived from those datasets, and following that the experimental protocol.

### 4.1 Datasets

#### 4.1.1 Cam-CAN dataset

Cam-CAN is the largest MEG dataset that is available, consisting of more than 600 subjects (Shafto et al., 2014). This MEG dataset is part of a much larger study that looks at the effects of aging, but at the time of this work only the second stage of the study had been released. In this stage subjects were involved in a clinical session with three MEG recordings, a resting state recording and a passive and active sensorimotor task. The same stimulus was used in both the active and passive stage which consisted of a checkerboard pattern or an auditory tone played at one of three different frequencies (300Hz, 600Hz and 1200Hz). These stimuli were presented for 34ms and 300ms for the visual and audio stimulus respectively. In the passive stage, which is used by Zubarev et al. (2019), no action is required by the subject and a trial consists of a unimodal stimulus, either auditory or visual, where half the trials are auditory and half are visual. This differs from both the active state and resting state recordings of the study which is not used in this work.

Our use of this dataset differs slightly from Zubarev et al. (2019). First, unlike Zubarev et al. (2019), we did not limit the number of subjects to 250. Instead, we only exclude five subjects from Cam-CAN (CC120208, CC510220, CC610462, CC620193, CC620685) where there were issues with the data (for example, sessions with no data), leaving 644 subjects and just over 77000 epochs. Second, partly as a consequence of this, we use a different split for training / validation / test, with our split chosen to allow us to run repeated experiments while still holding out a test set for use once only at the end, to genuinely measure model generalization. Specifically, we assign 60% of the subjects to the training set (388), 20% to the validation set (128), and the remaining 20% to the test set (128). In addition to the training, validation and test set, we also use a development set which is created by partitioning half the data (instead of subjects) from the validation set. We use the validation set for early stopping and hyperparameter selection, and the development set as a stand-in for the test set for producing results over the course of experiments. (Note that unlike the other splits, the validation-development split is not looking to evaluate inter-subject performance, but instead is used to evaluate intra-subject performance. This means that while the validation and test sets consist of different subjects, the validation and development have the same subjects but different trials.)

One other subtle difference concerns the epoching of the data, which in this context refers to dividing the continuous EEG signal into shorter, fixed intervals based on specific events, such as the onset of a stimulus. We broadly epoch the data in the same way as Zubarev et al. (2019), by taking an 800ms window that starts 300ms before the stimulus onset. However, in Zubarev et al. (2019), after the data was epoched it was downsampled and then normalized, whereas we do these final two steps in the reverse order. Zubarev et al.

Table 1: Literal translations of the Dutch sentences that were used as stimulus. Target words (highlighted in bold) are in the same ordinal position in both the Sentence and the Word List (in the original Dutch sentences)

|  | Sentence | Word List |
|---|---|---|
| Difficult | The nice lady gave Henk, who had bought a colourful **parrot**, a bag of seeds 
 *Het aardige vrouwtje gaf Henk die een kleurige **papegaai** gekocht had een zak pitjes* | Bag a colourful nice a had who lady **parrot** gave the bought seeds Henk 
 *Zak een kleurige aardige een had die vrouwtje **papegaai** gaf het gekocht pitjes Henk* |
| Easy | These are no regional **problems** such as those on the Antilles 
 *Dit zijn geen regionale **problemen** zoals die op de Antillen.* | such as no those Antilles **problems** regional are the these on 
 *zoals geen die Antillen **problemen** regionale zijn de dit op* |

(2019) normalized the data using the first 280ms of the downsampled data, leaving the rest to be processed by their neural networks. This means that this data would include a small amount of prestimulus data. In contrast, for strict correctness, we normalize based on the whole prestimulus period, and the input to the networks starts precisely at the stimulus onset.

### 4.1.2 Mother Of Unification Studies

In addition to Cam-CAN, another large dataset has also been recently released, the Mother Of Unification Studies (MOUS) (Schoffelen et al., 2019), which has a much richer stimulus. The focus of the study was understanding the way that humans process written and spoken language, with a focus on the processing of individual words in a sentence. The study consisted of 204 participants and like the Cam-CAN dataset, the subjects were exposed to an auditory and a visual stimulus giving us more than 50000 epochs in total. We split these subjects in the same manner as Cam-CAN, with 124 subjects in the training set, 40 in the validation set and 40 in the test set. In this case, however, subjects were only subjected to one stimulus, with half being shown written text and the other half hearing spoken words. In each case, the stimulus consists of linguistic utterances (in Dutch), either a valid sentence or an arbitrary list of words. In addition, there were two types of sentences: a sentence with a main clause plus a simple subordinate clause that is easier for cognitive processing, and a sentence with a relative clause that is harder to process. Sentences or word lists were presented word-by-word with a mean duration of 351 ms for each word (minimum of 300 ms and maximum of 1400 ms, depending on word length). Examples of the stimuli can be found in Table 1. In this work, we will look at three different classification tasks: distinguishing auditory from visual stimuli (AUDIOVIS), sentences from word lists (SENTWORDLIST), and sentences that are easy to process from those harder to process (HARDEASYSENT).

Our three classification tasks all use the same data. Our data is structured to align with the classification tasks used by (Zubarev et al., 2019), and as a result we restrict our window to contain only a single word. For each trial, we extract a window of the data from the point where the subject was presented with the target word. We epoch the data in the same way as Zubarev et al. (2019) as described in §4.1.1, taking 300ms before the onset and 500ms after. However, this results in a small change in the resulting input because the MOUS is sampled at a higher frequency. So while the final Cam-CAN data has 64 time samples, the final MOUS data has 85. In both cases we are normalizing by the mean and standard deviation of the first 36 samples of the epoched data which means that the MOUS data includes more prestimulus data.

The other deviation from the Cam-CAN data is the number of channels that are included. We are using 270 channels from the MOUS (compared to 204 for Cam-CAN), but we have also increased the number of time samples that are included as well. This does not constitute all of the channels that are available in the datasets: there are a number of channels which are not present in all recordings, and there were also channels from other sources (such as EEG). These channels were not considered in any of our experiments.

In total ten channels were not present in all recordings (BP2, EEG061, EEG062, EEG063, EEG064, MLC11, MLF62, MLT37, MRF66, MRO52); of these channels, 5 were MEG-related channels and were consequently excluded.

In terms of our derived classification tasks, the auditory vs visual stimulus task (AUDIOVIS) is similar to that for the Cam-CAN dataset in Zubarev et al. (2019): we are trying to predict if the subject was seeing a written word or if they were hearing a spoken word. However, it is important to note that unlike the Cam-CAN datasets this is a between-subject variable. A subject was shown either the words, or they heard them; none of the subjects experienced both. This is important for two reasons. First, different subjects might encode the relationships between the stimuli very differently; and second, deep learning is very good at picking up on unintended features (particularly noise) that correlates strongly with the output class: Ribeiro et al. (2016) demonstrated this for image classification systems, noting a classification task where the presence of snow in an image leads a model to predict that the class is 'wolf' rather than the correct 'dog'. In this case, we will have to consider the possibility that the neural network is picking up a characteristic of the session (such as background noise) instead of the stimulus; we analyse the results in light of this.

In our second classification task (SENTWORDLIST) we aim to distinguish a sentence from an arbitrary list of words: more specifically, if the stimulus target word is part of a syntactically correct sentence or part of an arbitrary list of words. Similarly, our third classification task (HARDEASYSENT) is to predict if the target word was part of a sentence with a syntactically complex structure or a simple one.

We can further break down each of these last two classification problems into more fine-grained tasks by also taking into account how the stimulus is presented. We evaluate each task by training models on three different subsets of the data; audio, visual and both together. This means that in each of the audio and visual subset we will be restricted to 102 subjects, although this restriction may allow models to focus on more fine-tuned features for each of the modes of stimulus.

## 4.2   Training and Evaluation

Aligning with previous work with similarly balanced datasets, our main metric is classification accuracy. In addition to this we look at the variability of the results in two ways. To characterise variability across subjects, we calculate a 95% confidence Wilson score interval for the classifier (Brown et al., 2001), which is based on a binomial assumption; the lower and upper bounds are indicated by LB, UB respectively. We report these confidence intervals in the main results. For additional analysis, to characterise the pairwise difference between models,[1] we report the bootstrap confidence interval (95%) using the percentile method (Cohen, 1995) for per-subject accuracy under each model. For each bootstrap iteration for a pair of models, we sample random combinations of subjects with replacement and compute the mean difference in subject accuracy for the sampled subjects. Using these sampled differences across many iterations, we generate a 95% bootstrap confidence interval employing the percentile method (Cohen, 1995). This allows us to distinguish between differences in models that are statistically significantly different from those that are not. We give further detail in Appendix A.

In each case, we trained three different models and then applied the model with the best validation loss to the test set. During all development we use the development set as a proxy for the test set. In addition to the results on the test set, we also report the validation accuracy of the last model that was trained.

We trained each model on a GPU with a batch size of 128, using the Adam gradient descent optimization algorithm (Kingma & Ba, 2015) with a learning rate of $10^{-3}$ which optimized the cross-entropy loss of each model. We used early stopping on the validation loss with a patience of 3.

### 4.2.1   Baselines

Our core baselines are LF-CNN and VAR-CNN from Zubarev et al. (2019). Like Zubarev et al. (2019), we also include high-performing computer vision models: GoogLeNet (Szegedy et al., 2015) and ResNet18

---

[1] We calculate bootstrap interval between each pair of models twice, comparing the difference between A and B and again between B and A.

(He et al., 2016).[2] These models are designed to process images with three colour channels, so we added a $1 \times 1$ convolution with three filters to form three "colour" channels where each "colour" can have different a "brightness" that is optimized by the model. We also considered the SCNN and Ra-SCNN models of Kostas et al. (2019), whose code is also helpfully available. However, their models do not fit our data.[3]

### 4.2.2   TimeAutoencoder

**Training**   Instead of using a next-token prediction, we focus on the traditional autoencoder training task where we train our model to reproduce an input from a compressed latent embedding. Specifically, we train our model to minimise the mean square error between the output to the decoder and the original input. This architecture builds on the ideas from our other models and relies on the Temporal Residual Block that we described in §3. However, unlike our other models, this architecture has no method of learning any spatial relationships at all. The intention here is that by limiting the types of interactions that it can model, it will be forced to learn low-level features that are more easily generalized in downstream tasks.

As the TimeAutoencoder only operates on temporal relationships, the latent embedding that it generates can be thought of as data that has been downsampled. We can also train the TimeAutoencoder on a larger input window to generate a latent representation that has the same dimensions as the inputs that we used in §4.1, allowing more temporal data to be incorporated. As training the autoencoder is significantly more computationally intensive than the other models, we only evaluate the architecture by training the TimeAutoencoder on MOUS with 128 time steps which will then halve the temporal dimension. We train models in five different configurations: two baselines which use the raw data, and three configurations which incorporate the encoder and differ in how the encoder is trained.

**Configurations**   We compare five different experimental configurations. The first two (Raw64, Raw128) will not use the TimeAutoencoder at all and instead use the raw data, with either 64 or 128-time steps. These will allow us to evaluate whether any performance increase is simply due to extra information that is present in the larger input. Of those that use the autoencoder, the Frozen configuration will use the latent representation of the encoder as part of a pre-processing step, and the parameters of the encoder will not be updated. In contrast, the Unfrozen will also use the latent representation but will be fine-tuned as part of training the classifier. The Uninitialized will use an encoder with randomly initialized parameters, and like Unfrozen the parameters will be updated while training the classifier. For each of these configurations, we will train four different model architectures (SERes, SETra, LF-CNN and VAR-CNN). We will apply these to the HardEasySent and SentWordlist tasks on MOUS, to assess the benefit of temporal information there.

Despite the simplicity of the TimeAutoencoder, there is a substantial computational expense and training these models took more than 8 hours. As a result, unlike our other experiments, we only train one instance of each model, do not develop stimulus-specific models, and use the Both subset to train the models. In addition, because of the increased memory requirements, we use a batch size of 32. To compensate for this we accumulate gradients across 4 batches to maintain an effective batch size of 128.

## 5   Results

In this section, we first present the performance on each task of our new models (TimeConv, SERes, SETra) relative to the selected baselines, followed our investigation on the utility of incorporating the TimeAutoencoder.

---

[2]Zubarev et al. (2019) used the older VGGNet, modified to include batch normalization. When we implemented it, we were unable to train a single model successfully. Instead of altering the architecture, we implemented architectures that already incorporated batch normalization.

[3]Their models assume a much larger temporal window than we are using. For instance, we found that when running their code for the BCI dataset, the temporal window is 1125. In contrast, we use 64 time samples, and their model cannot be run on a window of this size using the published configuration.

Table 2: Results of the AUDIOVIS task on both the Cam-CAN and MOUS datasets. Included is the Validation Accuracy, Test Subject Accuracy (mean ± 95% Wilson score CI) and the Lower and Upper Bound of the Subject Accuracy (i.e. bottom and top of the 95% CI). Highest results for each dataset are in bold.

| Dataset | Architecture | Val. Acc. | Test Sub. Acc. | Sub. Acc. LB | Sub. Acc. UB |
|---------|--------------|-----------|----------------|--------------|--------------|
| Cam-CAN | SERes (ours) | **95.12** | 95.70±0.8 | 94.93 | **96.47** |
| | TimeConv (ours) | 94.91 | **95.71±0.7** | **94.96** | 96.46 |
| | SETra (ours) | 93.48 | 94.31±1.0 | 93.34 | 95.28 |
| | LF-CNN | 93.19 | 93.87±1.1 | 92.80 | 94.95 |
| | VAR-CNN | 92.84 | 94.23±0.9 | 93.29 | 95.18 |
| | GoogLeNet | 93.10 | 94.05±0.9 | 93.14 | 94.96 |
| | ResNet18 | 94.09 | 94.32±1.0 | 93.36 | 95.27 |
| MOUS | SERes (ours) | 77.54 | 77.63±5.5 | 72.09 | 83.16 |
| | TimeConv (ours) | 74.80 | 74.03±5.1 | 68.89 | 79.17 |
| | SETra (ours) | 72.63 | 73.57±4.2 | 69.41 | 77.73 |
| | LF-CNN | 79.75 | 82.92±4.9 | 78.03 | 87.81 |
| | VAR-CNN | **81.62** | **83.67±5.0** | **78.64** | **88.71** |
| | GoogLeNet | 75.71 | 73.04±9.5 | 63.58 | 82.50 |
| | ResNet18 | 75.07 | 75.30±6.8 | 68.54 | 82.05 |

## 5.1 AudioVis

Table 2 shows the results for the AUDIOVIS task for both the Cam-CAN and MOUS datasets. We report the overall accuracy on the validation set and the mean subject accuracy and the confidence interval on the test set. For more detail, bootstrap confidence intervals are given in Appendix A.

Overall, the differences in results between the two datasets are substantial, despite conceptually being a similar task. Our new SERes model statistically significantly outperforms all others on Cam-CAN, whereas existing models VAR-CNN and LF-CNN are best on MOUS. The important distinction between the tasks on these datasets is that in MOUS it is a between-subject classification task, where each subject is only exposed to one stimulus. It is likely then that LF-CNN and VAR-CNN in particular are picking up on characteristics of the session rather than the essential properties of the MEG signal for the task. We also observe the much larger variation in the results for MOUS than Cam-CAN, seen in the differences between subject accuracy LB and UB (e.g. for SERes, 1.54 for Cam-CAN versus 11.09 for MOUS). In terms of existing computer vision models ResNet18 and GoogLeNet, performance is poor to moderate on both datasets; we note here also that these models would not always train successfully to convergence, on this task or on the others.

## 5.2 SentWordlist

Table 3 shows the results of the SENTWORDLIST task on the MOUS dataset (bootstrap confidence intervals in Table 8). SERes, the best performing in AUDIOVIS on Cam-CAN, outperformed the LF-CNN and VAR-CNN baselines in every case for the Visual and Both subsets. The computer vision models significantly more difficult to train and thus generally do not do well on this task, and the GoogLeNet in particular falls behind. However the ResNet18 shows the potential of more complex architectures on the visual subset.

As in the AUDIOVIS task, there is a substantial difference between the different subsets, with the Visual subset much more challenging. This may be due to the way that the information can be processed. When a word is presented visually the subject can immediately focus on any part of the word. For audio, on the other hand, the subject's attention will always initially be at the start of the word which may result in more informative temporal relationships. Despite having half the available data, the Audio subset achieves the best results.

Training on all of the data does not seem to improve compared to the results of each subset. This may be because the low-level features are not easily transferred from one modality to the other. This makes sense because these models are limited to learning a spatial embedding which only allows focusing on the brain activity in 16 distinct ways. When trained together, the models will need to focus on activity that applies on both modalities. Increasing the size of the spatial embedding could improve the results, although it may also make it easier for the network to overfit.

Table 3: Results of the SENTWORDLIST task on the MOUS dataset

| Subset | Architecture | Val. Acc. | Test Sub. Acc. | Sub. Acc. LB | Sub. Acc. UB |
|--------|--------------|-----------|----------------|--------------|--------------|
| Audio | SERes (ours) | **66.33** | **67.21±2.5** | **64.75** | **69.68** |
| | TimeConv (ours) | 63.72 | 64.91±1.8 | 63.07 | 66.74 |
| | SETra (ours) | 64.28 | 65.86±2.1 | 63.74 | 67.98 |
| | LF-CNN | 63.85 | 64.27±2.1 | 62.22 | 66.32 |
| | VAR-CNN | 62.83 | 64.47±2.4 | 62.11 | 66.82 |
| | GoogLeNet | 51.09 | 50.82±1.7 | 49.16 | 52.47 |
| | ResNet18 | 63.25 | 64.25±1.9 | 62.38 | 66.11 |
| Both | SERes (ours) | **58.61** | **59.99±2.1** | 57.84 | **62.14** |
| | TimeConv (ours) | 57.67 | 58.87±1.9 | 57.00 | 60.73 |
| | SETra (ours) | 58.21 | 59.86±1.9 | **57.93** | 61.79 |
| | LF-CNN | 57.60 | 58.32±1.8 | 56.57 | 60.07 |
| | VAR-CNN | 57.36 | 58.29±2.0 | 56.29 | 60.29 |
| | GoogLeNet | 52.64 | 52.91±1.1 | 51.83 | 53.98 |
| | ResNet18 | 56.56 | 58.46±1.7 | 56.81 | 60.12 |
| Visual | SERes (ours) | 54.97 | 56.60±1.4 | 55.20 | 58.00 |
| | TimeConv (ours) | **56.07** | 56.24±2.0 | 54.28 | 58.19 |
| | SETra (ours) | 55.96 | 56.18±1.5 | 54.69 | 57.68 |
| | LF-CNN | 55.96 | 55.16±1.7 | 53.49 | 56.82 |
| | VAR-CNN | 55.05 | 55.50±1.7 | 53.80 | 57.19 |
| | GoogLeNet | 53.72 | 55.10±0.6 | 54.50 | 55.69 |
| | ResNet18 | 53.57 | **57.33±1.6** | **55.76** | **58.91** |

### 5.3 HardEasySent

Table 4 shows the results of the HARDEASYSENT task on the MOUS dataset for both the Audio and Visual subsets (bootstrap confidence intervals in Table 9). Like the SENTWORDLIST task, there is a difference between the subsets, but in this case, the classes are more easily distinguishable with a Visual stimulus compared to the Audio equivalent. We also note that only one of the models (this same SETra) trained on the Audio subset did better than chance (50%), whereas in both of the other subsets each model lower bound of the 95% confidence interval was better than chance.

This also seems to be a task that does not see much benefit of the temporal features that our proposed models focus on. The best overall result is the SETra, with this Visual stimulus, although none of the results for models under the Visual stimulus are statistically significantly different. On the Both and Audio subsets the LF-CNN scores highest, but looking at statistical significance, only the computer vision architectures are actually worse on Both, and as noted above, most results on Audio are essentially at chance. In any case, this is a challenging task, but all of the models are able to distinguish the difference when the stimulus is presented visually.

### 5.4 TimeAutoencoder

Table 5 shows the results of applying TimeAutoencoder to LF-CNN and VAR-CNN on the SENTWORDLIST task, which have substantially improved relative to other models. (Table 10 gives the bootstrap confidence

Table 4: Results of the HARDEASYSENT task on the MOUS dataset

| Subset | Architecture | Val. Acc. | Test Sub. Acc. | Sub. Acc. LB | Sub. Acc. UB |
|--------|--------------|-----------|----------------|--------------|--------------|
| Audio | SERes (ours) | 50.79 | 49.95±0.4 | 49.50 | 50.40 |
| | TimeConv (ours) | 50.71 | 49.70±0.7 | 49.00 | 50.39 |
| | SETra (ours) | **51.95** | 51.83±1.3 | **50.54** | 53.12 |
| | LF-CNN | 51.70 | **51.92±2.3** | 49.59 | **54.25** |
| | VAR-CNN | 50.04 | 49.96±1.8 | 48.16 | 51.75 |
| | GoogLeNet | 51.79 | 49.17±1.1 | 48.10 | 50.25 |
| | ResNet18 | 51.79 | 50.02±1.6 | 48.46 | 51.59 |
| Both | SERes (ours) | 54.11 | 54.97±1.6 | 53.36 | 56.58 |
| | TimeConv (ours) | 54.60 | 54.90±1.8 | 53.13 | 56.67 |
| | SETra (ours) | 53.81 | 55.26±1.7 | 53.56 | 56.97 |
| | LF-CNN | 54.49 | **56.02±1.5** | **54.56** | **57.49** |
| | VAR-CNN | **55.89** | 55.34±1.6 | 53.70 | 56.98 |
| | GoogLeNet | 53.62 | 53.29±1.3 | 51.94 | 54.64 |
| | ResNet18 | 55.06 | 53.51±1.9 | 51.64 | 55.37 |
| Visual | SERes (ours) | 57.27 | 59.78±1.6 | 58.19 | 61.37 |
| | TimeConv (ours) | 55.25 | 58.20±2.0 | 56.17 | 60.24 |
| | SETra (ours) | **57.90** | **59.98±1.7** | 58.27 | **61.69** |
| | LF-CNN | 56.37 | 59.40±1.1 | 58.35 | 60.46 |
| | VAR-CNN | 56.44 | 59.24±2.2 | 57.04 | 61.44 |
| | GoogLeNet | 57.83 | 59.16±0.6 | **58.60** | 59.72 |
| | ResNet18 | 57.62 | 58.58±0.9 | 57.66 | 59.50 |

intervals for comparing across encoders, and Table 12 for comparing across models using the best encoder.) The best configuration (UNFROZEN) produced Test Accuracies of 59.56% and 59.84%, respectively. This reduced the difference in test accuracy between SERes and VAR-CNN to just 0.53% from 2.08%. For both the LF-CNN and VAR-CNN the UNFROZEN configuration outperforms all other configuration in every evaluation set.

Interestingly, the FROZEN in contrast decreases performance in most cases. However, we can also see the benefit of pre-training by comparing the results to UNINITIALIZED, which are consistently lower on the LF-CNN and VAR-CNN models. This might suggest that the TimeAutoencoder is overfitting and is learning how to model noise as well as useful information.

Looking at SERes and SETra, Disabled (128) performs better than the Disabled (64) in three out or four configurations, suggesting that unlike LF-CNN and VAR-CNN, these models are more capable of exploiting temporal information. We also see that these models perform no better when the encoder is added. Taken together this suggests that the these models are already sufficiently capable of modelling the temporal relationships; and that the potential overfitting from the pretrained network may outweigh the benefit to further temporal modelling.

Table 6, containing results on the HARDEASYSENT task, shows no clear optimal configuration. (Table 11 gives the bootstrap confidence intervals for comparing across encoders, and Table 13 for comparing across models using the best encoder.) This is not entirely surprising: we observed in Table 4 and §5.3 that the models that incorporated temporal features performed no better than those without. This result more directly supports the idea that temporal features are more important in some tasks than others.

## 6 Discussion

We have introduced three novel architectures that have demonstrated that, by learning temporal relationships in the data, we are able to improve the performance on temporally oriented tasks. In particular, our SERes

Table 5: Results of the applying the TimeAutoencoder to the SENTWORDLIST task on the MOUS dataset

| Architecture | Encoder | Val. Acc. | Test Sub. Acc. | Sub. Acc. LB | Sub. Acc. UB |
|---|---|---|---|---|---|
| SERes (ours) | Disabled (128) | **59.94** | **60.37±2.2** | **58.18** | **62.55** |
| | Disabled (64) | 58.13 | 59.34±2.1 | 57.25 | 61.43 |
| | Frozen | 58.61 | 59.21±1.9 | 57.34 | 61.08 |
| | Unfrozen | 57.69 | 59.18±1.9 | 57.25 | 61.10 |
| | Uninitialized | 59.37 | 59.82±2.0 | 57.86 | 61.78 |
| SETra (ours) | Disabled (128) | 59.11 | 60.26±2.0 | **58.29** | 62.24 |
| | Disabled (64) | 58.21 | 59.59±2.0 | 57.58 | 61.60 |
| | Frozen | 55.92 | 57.21±1.5 | 55.73 | 58.69 |
| | Unfrozen | 57.93 | 58.32±2.0 | 56.29 | 60.35 |
| | Uninitialized | **59.57** | 60.28±2.1 | 58.18 | **62.38** |
| LF-CNN | Disabled (128) | 57.89 | 58.61±1.9 | 56.67 | 60.55 |
| | Disabled (64) | 57.95 | 58.69±2.0 | 56.73 | 60.65 |
| | Frozen | 55.98 | 56.85±1.7 | 55.19 | 58.51 |
| | Unfrozen | **59.11** | **59.56±1.7** | **57.83** | **61.29** |
| | Uninitialized | 58.83 | 59.30±1.8 | 57.49 | 61.10 |
| VAR-CNN | Disabled (128) | 57.06 | 58.29±2.0 | 56.30 | 60.27 |
| | Disabled (64) | 57.69 | 58.36±1.9 | 56.42 | 60.31 |
| | Frozen | 55.03 | 55.88±1.4 | 54.46 | 57.30 |
| | Unfrozen | 58.85 | **59.84±2.1** | **57.71** | **61.97** |
| | Uninitialized | **58.99** | 59.56±1.9 | 57.64 | 61.48 |

Table 6: Results of the applying the TimeAutoencoder to the HARDEASYSENT task on the MOUS dataset

| Architecture | Encoder | Val. Acc. | Test Sub. Acc. | Sub. Acc. LB | Sub. Acc. UB |
|---|---|---|---|---|---|
| SERes (ours) | Disabled (128) | 54.07 | 53.98±1.4 | 52.54 | 55.42 |
| | Disabled (64) | 53.20 | **54.68±1.7** | **52.99** | **56.38** |
| | Frozen | **54.11** | 54.35±1.5 | 52.84 | 55.86 |
| | Unfrozen | **54.11** | 54.35±1.5 | 52.84 | 55.86 |
| | Uninitialized | **54.11** | 54.35±1.5 | 52.84 | 55.86 |
| SETra (ours) | Disabled (128) | 54.60 | **55.84±1.7** | **54.10** | **57.58** |
| | Disabled (64) | 53.13 | 54.41±1.6 | 52.80 | 56.01 |
| | Frozen | 54.38 | 54.24±1.5 | 52.75 | 55.74 |
| | Unfrozen | 54.11 | 54.33±1.5 | 52.83 | 55.83 |
| | Uninitialized | **55.25** | 54.69±1.9 | 52.81 | 56.57 |
| LF-CNN | Disabled (128) | 54.11 | 55.34±1.5 | 53.80 | 56.88 |
| | Disabled (64) | 52.75 | 55.56±1.6 | 54.01 | 57.12 |
| | Frozen | 53.92 | 54.43±1.3 | 53.15 | 55.72 |
| | Unfrozen | 54.26 | **55.90±1.7** | **54.18** | **57.63** |
| | Uninitialized | **54.41** | 54.57±1.8 | 52.81 | 56.33 |
| VAR-CNN | Disabled (128) | 53.62 | 54.86±2.0 | 52.91 | 56.81 |
| | Disabled (64) | 53.96 | 54.79±1.9 | 52.94 | 56.64 |
| | Frozen | 52.29 | 51.35±1.6 | 49.73 | 52.97 |
| | Unfrozen | **55.17** | **55.62±1.8** | **53.80** | **57.44** |
| | Uninitialized | 53.77 | 54.43±1.5 | 52.94 | 55.91 |

was able to beat all our baseline models on the SENTWORDLIST task where it achieved the highest overall accuracy in two of three subsets (Audio and Both) and was only outperformed by our ResNet18 in the third

(Visual) subset. This is the task that is most strongly temporally oriented: what distinguishes a grammatical sentence from an arbitrarily ordered list of those same words is exactly the temporal sequence of the words. Supporting that, at the level of an individual word (with the stimulus being presented one word at a time), we point out that these temporal patterns were easiest to identify on the audio subset of the data. This difference may be due to the way that the word is processed because the subject's attention will always initially be at the start of the word for the audio stimulus whereas this is not the case for the visual stimulus. (We note also that it was on this task where the TimeAutoencoder boosted baseline model scores.)

Similarly, SERes performs better by a statistically significant margin than all other models except TimeConv on the AudioVis task of the Cam-CAN dataset. In both of these tasks, as well as performing better than the simpler LF-CNN and VAR-CNN models, it beats the computer vision baselines even though these have more than 150 times more trainable parameters.

On the other hand, where the simpler LF-CNN and VAR-CNN models do do significantly better than all of our new models, on the AudioVis task on the MOUS dataset, this is likely due to learning characteristics of the session, because of the between-subjects nature of the experimental data. Less definitively, in the HardEasySent task, our SETra performs well as does LF-CNN and VAR-CNN, although the difference between models is not statistically significant with the exception of SETra, which is the only model to do better than chance on the visual subset, and the computer vision models which significantly underperform on the both subset. It is interesting to consider how this HardEasySent might differ from SentWordlist. We note first that, following Zubarev et al. (2019), we only look at a very short time interval (the 500ms period immediately after onset); with the mean duration of word utterance being 351ms as well as a 300ms delay between words this period only includes one word. It has been recognised for some time that relative clauses of the sort in the MOUS dataset are cognitively challenging (e.g. Bach et al. (1986)) — unlike main clauses where the verb comes second (e.g. *gaf* 'gave' in the first example of Table 1), in these relative clauses the tensed verb comes at the end (e.g. *had* 'had' in that same example). Work such as Bach et al. (1986) has theorised that in these kinds of relative clause constructions, noun phrases in the relative clause must be stored in memory awaiting assignment to their respective verbs that appear at the end of the clause, although the precise details of the temporal course of lexical processing still constitute an open question (e.g. de Goede et al. (2009)). That is, what differentiates the harder from simpler constructions in the dataset in terms of processing complexity might not occur until after the time interval we use. Within that interval, all that differentiates the types of sentences is whether a relative pronoun (e.g. *die* 'who', 'that', etc) or other lexical item (e.g. *zoals* 'such as') is used. That is, the task may rely more on distinguishing among single lexical items than any strongly temporal aspect.

## 7 Conclusion

In this paper we introduced three new deep learning models for MEG data, and evaluated them on a previously defined large-dataset classification task, as well as several new classification tasks derived from the recently released MOUS dataset. Compared with prior work, the new classification tasks demonstrate a wide range of difficulty, allowing differentiation of the performance of models.

Our three new architectures concentrated on learning temporal relationships in the data that were ignored in the prior literature. Our new SERes model outperformed both previous state of the art and baseline computer vision architectures on the previously defined classification task on the Cam-CAN dataset, as well as the new tasks where a temporal aspect was clear. We further found that an autoencoder-based preprocessing component that learned temporal information could be combined with previous models to improve their performance on temporal tasks.

MEG is a relatively new domain for deep learning, and there is much to be understood on the best way to train models. We have noted that models which contain more than one million parameters (GoogLeNet and ResNet18) would not always train successfully. This suggests that there is some unknown, intermittent and undesirable interaction taking place while training these models. Understanding the cause of this may increase the performance of all architectures significantly.

In this work we have shown the benefits of pre-training using an autoencoder; however, we constrained the encoder to only be able to learn temporal features. Developing a method of pre-training that builds on spatial features as well is likely to further increase performance.

In terms of characterising what kinds of tasks will be improved by including temporal information, while our SERes architecture achieved state-of-the-art results on temporally-oriented tasks, it is not clear what aspects of those tasks benefits from the architecture. A deeper understanding of the underlying cognitive processes that cause the difference in temporal behavior between our tasks derived from the MOUS dataset is necessary; this kind of understanding will likely contribute to further performance improvements in neural network architectures for MEG processing.

Another important direction for future research is the visualisation of the activation patterns. Zubarev et al. (2019) used traditional source estimation algorithms (Dale et al., 2000) in a way that was fairly straightforward for their particular relatively shallow architectures but that might be more complex for our models; the deep learning-based source estimation work of Pantazis & Adler (2021) that has appeared since that time is a promising alternative for integrating into our models.

In summary, we have made four main contributions. First, we have introduced three temporally rich deep learning models. Second, we have evaluated these models to existing work on both existing and novel more challenging classification tasks. Third, we have demonstrated that our proposed models outperform the previous state-of-the-art on temporally-oriented classification tasks. Finally, we have shown that when we incorporate a temporal preprocessing component into existing models, we can improve their performance on temporal tasks.

Broader implications of this work span both methodological advancements and applications in neuroscience. Deep learning's integration into Magnetoencephalography (MEG) data analysis, as showcased here, underscores a paradigm shift in our approach to neuroimaging. The ability to identify nuanced patterns using intricate deep neural network architectures could pave the way for more accurate diagnoses and improved understanding of neurological disorders. As the intersection of neuroscience and machine learning continues to grow, the contributions of this study could serve as a foundational step for subsequent research endeavors, driving the pursuit of better insights into cognitive processes and brain functions.

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

# A  Pairwise bootstrap confidence intervals

This section contains the results of the pairwise bootstrap confidence intervals for the models. For convenience, we repeat the description of the method we use to compute the confidence intervals.

These statistical tests look at the pairwise difference in subject accuracy between models and we report the bootstrap confidence interval (95%) using the percentile method (Cohen, 1995) for per-subject accuracy under each model. For each bootstrap iteration for a pair of models, we sample random combinations of subjects with replacement and compute the mean difference in subject accuracy for the sampled subjects. Using these sampled differences across many iterations, we generate a 95% bootstrap confidence interval employing the percentile method (Cohen, 1995). This allows us to distinguish between differences in models that are statistically significantly different from those that are not.

We repeat the process twice for each pair of models, comparing the difference between A and B and again between B and A. For each test, we report the lower and upper bounds of the confidence interval. If the interval covers 0 they are not statistically significantly different; if the lower bound is positive, the model in the row is significantly better than the one in the column. If the upper bound is negative, the model in the column is significantly better than the one in the row. Statistically significant differences are in bold. So, for example, in Table 7, the SERes model is statistically significantly better than all others except TimeConv.

The tables in this appendix present pairwise bootstrap confidence intervals for all datasets and tasks in the main paper: Table 7 for AUDIOVIS, Table 8 for SENTWORDLIST, Table 9 for HARDEASYSENT, Table 10 for SENTWORDLIST autoencoder models, Table 11 for HARDEASYSENT autoencoder models, Table 12 for SENTWORDLIST autoencoder configurations, Table 13 for HARDEASYSENT autoencoder configurations.

Table 7: Pairwise bootstrap confidence intervals for models on the AUDIOVIS task. Statistically significant differences in bold.

| Subset | Architecture | SERes | TimeConv | SETra | LF-CNN | VAR-CNN | ResNet18 | GoogLeNet |
|---|---|---|---|---|---|---|---|---|
| Cam-CAN | SERes (ours) | - | (-0.36,0.35) | **(0.99,1.80)** | **(1.24,2.44)** | **(0.92,2.05)** | **(0.92,1.85)** | **(1.24,2.08)** |
| | TimeConv (ours) | (-0.33,0.36) | - | **(0.96,1.86)** | **(1.25,2.47)** | **(0.97,1.99)** | **(0.96,1.84)** | **(1.28,2.06)** |
| | SETra (ours) | **(-1.80,-0.99)** | **(-1.86,-0.96)** | - | **(0.03,0.87)** | (-0.39,0.55) | (-0.48,0.46) | (-0.21,0.72) |
| | LF-CNN | **(-2.44,-1.24)** | **(-2.45,-1.25)** | **(-0.85,-0.01)** | - | (-0.86,0.14) | (-0.96,0.07) | (-0.79,0.42) |
| | VAR-CNN | **(-2.03,-0.92)** | **(-2.01,-0.98)** | (-0.55,0.39) | (-0.14,0.87) | - | (-0.64,0.44) | (-0.40,0.77) |
| | GoogLeNet | **(-2.06,-1.25)** | **(-2.05,-1.27)** | (-0.73,0.21) | (-0.44,0.78) | (-0.76,0.42) | (-0.66,0.13) | - |
| | ResNet18 | **(-1.86,-0.92)** | **(-1.83,-0.96)** | (-0.47,0.49) | (-0.08,0.98) | (-0.45,0.63) | - | (-0.13,0.67) |
| MOUS | SERes (ours) | - | (-0.36,7.80) | (-0.01,7.94) | **(-8.33,-2.45)** | **(-8.17,-4.10)** | (-1.87,6.51) | (-0.13,10.07) |
| | TimeConv (ours) | (-7.75,0.51) | - | (-2.93,3.89) | **(-12.49,-5.34)** | **(-13.36,-6.15)** | (-4.64,2.36) | (-5.00,7.48) |
| | SETra (ours) | **(-8.00,-0.12)** | (-3.89,2.89) | - | **(-13.68,-4.75)** | **(-13.93,-6.31)** | (-6.83,3.40) | (-6.18,8.01) |
| | LF-CNN | **(2.46,8.32)** | **(5.34,12.45)** | **(4.91,13.66)** | - | (-2.64,1.08) | **(3.98,11.71)** | **(4.05,16.08)** |
| | VAR-CNN | **(4.11,8.09)** | **(6.04,13.27)** | **(6.33,13.77)** | (-1.11,2.62) | - | **(4.34,12.63)** | **(5.27,16.66)** |
| | GoogLeNet | **(-9.87,-0.00)** | (-7.47,5.17) | (-8.07,6.14) | **(-16.13,-4.27)** | **(-16.65,-5.36)** | (-7.21,2.58) | - |
| | ResNet18 | (-6.57,1.93) | (-2.32,4.62) | (-3.43,6.80) | **(-11.64,-3.83)** | **(-12.63,-4.43)** | - | (-2.69,7.36) |

Table 8: Pairwise bootstrap confidence intervals for models on the SENTWORDLIST task. Statistically significant differences in bold.

| Subset | Architecture | SERes | TimeConv | SETra | LF-CNN | VAR-CNN | ResNet18 | GoogLeNet |
|---|---|---|---|---|---|---|---|---|
| Audio | SERes (ours) | - | **(0.92,3.65)** | **(0.12,2.62)** | **(1.68,4.27)** | **(1.28,4.31)** | **(1.37,4.50)** | **(13.41,19.05)** |
| | TimeConv (ours) | **(-3.64,-0.90)** | - | (-2.51,0.54) | (-0.95,2.23) | (-1.02,1.95) | (-0.58,2.03) | **(11.73,16.16)** |
| | SETra (ours) | **(-2.61,-0.11)** | (-0.63,2.47) | - | **(-2.89,-0.37)** | (-0.08,2.77) | **(0.01,3.22)** | **(12.60,17.29)** |
| | LF-CNN | **(-4.27,-1.68)** | (-2.24,0.92) | **(0.39,2.89)** | - | (-1.84,1.48) | (-1.38,1.42) | **(10.89,15.85)** |
| | VAR-CNN | **(-4.31,-1.31)** | (-1.92,1.04) | (-2.78,0.10) | (-1.45,1.84) | - | (-1.14,1.51) | **(10.94,16.13)** |
| | GoogLeNet | **(-19.12,-13.47)** | **(-16.13,-11.75)** | **(-17.31,-12.61)** | **(-15.83,-10.84)** | **(-16.20,-10.91)** | **(-15.44,-11.31)** | - |
| | ResNet18 | **(-4.49,-1.45)** | (-2.04,0.59) | **(-3.22,-0.01)** | (-1.41,1.36) | (-1.55,1.19) | - | **(11.34,15.47)** |
| Both | SERes (ours) | - | **(0.15,2.13)** | (-0.72,0.97) | **(0.52,2.85)** | **(0.20,3.30)** | **(0.60,2.52)** | **(4.42,9.74)** |
| | TimeConv (ours) | **(-2.13,-0.16)** | - | (-2.04,0.09) | (-0.46,1.63) | (-0.90,1.95) | (-0.46,1.25) | **(3.56,8.32)** |
| | SETra (ours) | (-0.97,0.71) | (-0.08,2.03) | - | **(0.50,2.58)** | **(0.42,2.74)** | **(0.37,2.35)** | **(4.59,9.36)** |
| | LF-CNN | **(-2.83,-0.50)** | (-1.65,0.47) | **(-2.58,-0.46)** | - | (-1.32,1.38) | (-1.25,0.94) | **(3.20,7.68)** |
| | VAR-CNN | **(-3.27,-0.22)** | (-1.95,0.87) | **(-2.73,-0.42)** | (-1.40,1.34) | - | (-1.50,1.13) | **(3.13,7.67)** |
| | GoogLeNet | **(-9.71,-4.46)** | **(-8.32,-3.62)** | **(-9.38,-4.65)** | **(-7.71,-3.15)** | **(-7.79,-3.17)** | **(-7.71,-3.44)** | - |
| | ResNet18 | **(-2.55,-0.58)** | (-1.27,0.46) | **(-2.35,-0.38)** | (-0.93,1.21) | (-1.14,1.46) | - | **(3.41,7.71)** |
| Visual | SERes (ours) | - | (-1.19,1.98) | (-0.83,1.69) | **(0.05,2.96)** | (-0.18,2.36) | (-2.15,0.70) | **(0.07,2.90)** |
| | TimeConv (ours) | (-2.00,1.16) | - | (-1.54,1.65) | (-0.56,2.79) | (-0.86,2.37) | (-2.55,0.45) | (-0.63,2.92) |
| | SETra (ours) | (-1.67,0.81) | (-1.68,1.54) | - | (-0.65,2.65) | (-0.80,2.19) | **(-2.25,-0.09)** | (-0.29,2.52) |
| | LF-CNN | (-2.91,0.03) | (-2.80,0.61) | (-2.64,0.64) | - | (-2.20,1.42) | **(-3.72,-0.67)** | (-1.63,1.55) |
| | VAR-CNN | (-2.38,0.16) | (-2.39,0.88) | (-2.19,0.80) | (-1.40,2.20) | - | **(-3.29,-0.20)** | (-1.01,1.79) |
| | GoogLeNet | **(-2.89,-0.08)** | (-2.89,0.62) | (-2.51,0.31) | (-1.54,1.65) | (-1.85,0.95) | **(-3.64,-0.76)** | - |
| | ResNet18 | (-0.71,2.13) | (-0.44,2.54) | **(0.09,2.23)** | **(0.70,3.72)** | **(0.24,3.31)** | - | **(0.80,3.64)** |

Table 9: Pairwise bootstrap confidence intervals for models on the HARDEASYSENT task. Statistically significant differences in bold.

| Subset | Architecture | SERes | TimeConv | SETra | LF-CNN | VAR-CNN | ResNet18 | GoogLeNet |
|---|---|---|---|---|---|---|---|---|
| Audio | SERes (ours) | - | (-0.60,1.00) | **(-2.86,-0.84)** | (-3.89,0.06) | (-1.54,1.52) | (-1.51,1.42) | (-0.28,1.83) |
| | TimeConv (ours) | (-1.04,0.60) | - | **(-3.44,-0.79)** | (-4.37,0.10) | (-1.99,1.56) | (-1.94,1.20) | (-0.52,1.74) |
| | SETra (ours) | **(0.80,2.83)** | **(0.75,3.43)** | - | (-2.14,2.09) | **(0.25,3.58)** | (-0.16,3.78) | **(1.01,4.30)** |
| | LF-CNN | (-0.07,3.94) | (-0.09,4.40) | (-2.06,2.15) | - | **(0.44,3.57)** | (-0.77,4.71) | **(0.37,5.08)** |
| | VAR-CNN | (-1.55,1.48) | (-1.52,1.96) | **(-3.58,-0.21)** | **(-3.57,-0.43)** | - | (-2.46,2.43) | (-1.28,2.89) |
| | GoogLeNet | (-1.83,0.29) | (-1.69,0.54) | **(-4.28,-1.07)** | **(-5.02,-0.40)** | (-2.88,1.29) | (-2.53,0.76) | - |
| | ResNet18 | (-1.40,1.52) | (-1.21,1.94) | (-3.75,0.16) | (-4.71,0.89) | (-2.44,2.43) | - | (-0.78,2.54) |
| Both | SERes (ours) | - | (-1.38,1.60) | (-1.68,1.07) | (-2.42,0.25) | (-1.65,0.86) | **(0.12,2.75)** | **(0.47,2.91)** |
| | TimeConv (ours) | (-1.53,1.37) | - | (-1.79,1.03) | (-2.72,0.45) | (-2.16,1.27) | (-0.36,3.16) | **(0.05,3.16)** |
| | SETra (ours) | (-1.10,1.68) | (-1.03,1.75) | - | (-2.01,0.46) | (-1.47,1.23) | **(0.16,3.35)** | **(0.61,3.26)** |
| | LF-CNN | (-0.29,2.39) | (-0.41,2.74) | (-0.47,2.05) | - | (-0.58,1.96) | **(1.21,3.96)** | **(1.59,3.83)** |
| | VAR-CNN | (-0.86,1.63) | (-1.23,2.14) | (-1.23,1.48) | (-1.93,0.58) | - | **(0.66,3.04)** | **(0.92,3.20)** |
| | GoogLeNet | **(-2.89,-0.47)** | **(-3.16,-0.05)** | **(-3.30,-0.63)** | **(-3.84,-1.58)** | **(-3.21,-0.91)** | (-1.39,1.04) | - |
| | ResNet18 | **(-2.80,-0.17)** | (-3.16,0.37) | **(-3.29,-0.20)** | **(-3.95,-1.18)** | **(-3.02,-0.66)** | - | (-1.02,1.39) |
| Visual | SERes (ours) | - | (-0.48,3.54) | (-2.15,1.47) | (-1.45,2.04) | (-1.84,2.53) | (-0.29,2.75) | (-0.79,2.01) |
| | TimeConv (ours) | (-3.53,0.48) | - | (-3.63,0.16) | (-3.22,0.78) | (-2.63,0.67) | (-1.92,1.11) | (-2.64,0.66) |
| | SETra (ours) | (-1.47,2.04) | (-0.10,3.62) | - | (-0.94,2.11) | (-1.40,2.63) | (-0.37,3.14) | (-0.77,2.49) |
| | LF-CNN | (-2.04,1.38) | (-0.83,3.22) | (-2.13,0.97) | - | (-1.91,2.04) | (-0.31,1.97) | (-0.96,1.31) |
| | VAR-CNN | (-2.49,1.89) | (-0.62,2.62) | (-2.61,1.41) | (-2.03,1.92) | - | (-1.26,2.70) | (-1.75,2.13) |
| | GoogLeNet | (-2.01,0.79) | (-0.72,2.66) | (-2.51,0.79) | (-1.31,0.93) | (-2.10,1.75) | (-0.03,1.23) | - |
| | ResNet18 | (-2.74,0.32) | (-1.13,1.92) | (-3.21,0.35) | (-1.96,0.30) | (-2.72,1.25) | - | (-1.23,0.01) |

Table 10: Pairwise bootstrap confidence intervals for models with the best performing encoders configuration on the SENTWORDLIST task. Statistically significant differences in bold.

| | SERes | SETra | LF-CNN | VAR-CNN |
|---|---|---|---|---|
| SERes (ours) | - | (-0.79,0.97) | (-0.19,1.82) | (-0.61,1.67) |
| SETra (ours) | (-0.97,0.79) | - | (-0.14,1.54) | (-0.57,1.40) |
| LF-CNN | (-1.85,0.19) | (-1.54,0.16) | - | (-1.24,0.76) |
| VAR-CNN | (-1.73,0.61) | (-1.41,0.53) | (-0.74,1.26) | - |

Table 11: Pairwise bootstrap confidence intervals for models with the best performing encoders configuration on the HARDEASYSENT task. Statistically significant differences in bold.

| | SERes | SETra | LF-CNN | VAR-CNN |
|---|---|---|---|---|
| SERes (ours) | - | (-2.43,0.13) | (-2.90,0.52) | (-2.32,0.49) |
| SETra (ours) | (-0.12,2.45) | - | (-1.42,1.28) | (-1.10,1.57) |
| LF-CNN | (-0.55,2.93) | (-1.29,1.41) | - | (-1.18,1.77) |
| VAR-CNN | (-0.53,2.33) | (-1.61,1.07) | (-1.78,1.15) | - |

Table 12: Pairwise bootstrap confidence intervals for encoders configurations of models on the SENT-WORDLIST task. Statistically significant differences in bold.

| Model | Encoder Encoder | Disabled (128) | Disabled (64) | Frozen | Unfrozen | Uninitialized |
|---|---|---|---|---|---|---|
| SERes (ours) | Disabled (128) | - | (-0.08,2.11) | **(0.24,2.06)** | **(0.17,2.25)** | (-0.38,1.48) |
| | Disabled (64) | (-2.12,0.07) | - | (-0.76,0.99) | (-0.85,1.25) | (-1.73,0.73) |
| | Frozen | **(-2.05,-0.23)** | (-1.00,0.74) | - | (-0.84,0.92) | (-1.47,0.21) |
| | Unfrozen | **(-2.22,-0.18)** | (-1.24,0.88) | (-0.93,0.81) | - | (-1.84,0.57) |
| | Uninitialized | (-1.50,0.39) | (-0.69,1.70) | (-0.21,1.47) | (-0.57,1.85) | - |
| SETra (ours) | Disabled (128) | - | (-0.10,1.41) | **(1.61,4.45)** | **(1.02,2.90)** | (-0.72,0.68) |
| | Disabled (64) | (-1.44,0.07) | - | **(1.14,3.65)** | **(0.31,2.29)** | (-1.64,0.28) |
| | Frozen | **(-4.46,-1.66)** | **(-3.65,-1.17)** | - | **(-2.23,-0.03)** | **(-4.68,-1.49)** |
| | Unfrozen | **(-2.89,-1.01)** | **(-2.28,-0.29)** | **(0.04,2.24)** | - | **(-2.96,-0.97)** |
| | Uninitialized | (-0.69,0.72) | (-0.28,1.64) | **(1.49,4.67)** | **(0.98,2.96)** | - |
| LF-CNN | Disabled (128) | - | (-1.15,0.97) | **(0.85,2.73)** | (-2.08,0.21) | (-1.95,0.54) |
| | Disabled (64) | (-0.98,1.17) | - | **(0.79,2.93)** | (-1.78,0.06) | (-1.65,0.52) |
| | Frozen | **(-2.72,-0.85)** | **(-2.97,-0.80)** | - | **(-3.63,-1.83)** | **(-3.55,-1.36)** |
| | Unfrozen | (-0.20,2.09) | (-0.04,1.78) | **(1.83,3.62)** | - | (-0.64,1.18) |
| | Uninitialized | (-0.54,1.94) | (-0.49,1.68) | **(1.38,3.51)** | (-1.22,0.62) | - |
| VAR-CNN | Disabled (128) | - | (-1.29,1.13) | **(1.08,3.76)** | **(-2.84,-0.23)** | (-2.56,0.03) |
| | Disabled (64) | (-1.15,1.30) | - | **(0.94,4.06)** | **(-2.56,-0.40)** | **(-2.16,-0.27)** |
| | Frozen | **(-3.72,-1.10)** | **(-4.06,-0.97)** | - | **(-5.55,-2.34)** | **(-5.17,-2.21)** |
| | Unfrozen | **(0.21,2.85)** | **(0.38,2.56)** | **(2.38,5.57)** | - | (-0.66,1.20) |
| | Uninitialized | **(0.01,2.56)** | **(0.25,2.15)** | **(2.24,5.16)** | (-1.19,0.65) | - |

Table 13: Pairwise bootstrap confidence intervals for encoders configurations of models on the HARDEASY-SENT task. Statistically significant differences in bold.

| Model | Encoder / Encoder | Disabled (128) | Disabled (64) | Frozen | Unfrozen | Uninitialized |
|---|---|---|---|---|---|---|
| SERes (ours) | Disabled (128) | - | (-1.76,0.36) | (-1.05,0.29) | (-1.05,0.28) | (-1.05,0.28) |
| | Disabled (64) | (-0.35,1.80) | - | (-0.58,1.29) | (-0.61,1.28) | (-0.61,1.25) |
| | Frozen | (-0.28,1.04) | (-1.23,0.58) | - | - | - |
| | Unfrozen | (-0.27,1.06) | (-1.28,0.60) | - | - | - |
| | Uninitialized | (-0.29,1.05) | (-1.28,0.60) | - | - | - |
| SETra (ours) | Disabled (128) | - | **(0.09,2.86)** | **(0.48,2.75)** | **(0.38,2.69)** | (-0.14,2.48) |
| | Disabled (64) | **(-2.83,-0.08)** | - | (-1.10,1.41) | (-1.22,1.34) | (-1.70,1.12) |
| | Frozen | **(-2.74,-0.50)** | (-1.39,1.11) | - | (-0.33,0.15) | (-1.64,0.76) |
| | Unfrozen | **(-2.70,-0.40)** | (-1.35,1.21) | (-0.15,0.33) | - | (-1.51,0.81) |
| | Uninitialized | (-2.51,0.14) | (-1.13,1.67) | (-0.78,1.64) | (-0.79,1.50) | - |
| LF-CNN | Disabled (128) | - | (-1.49,1.07) | (-0.55,2.32) | (-1.90,0.78) | (-0.90,2.28) |
| | Disabled (64) | (-1.09,1.51) | - | (-0.31,2.55) | (-1.82,1.08) | (-0.61,2.52) |
| | Frozen | (-2.36,0.55) | (-2.52,0.31) | - | **(-2.64,-0.26)** | (-1.62,1.39) |
| | Unfrozen | (-0.81,1.90) | (-1.08,1.77) | **(0.23,2.66)** | - | (-0.33,2.91) |
| | Uninitialized | (-2.33,0.86) | (-2.54,0.58) | (-1.35,1.64) | (-2.93,0.36) | - |
| VAR-CNN | Disabled (128) | - | (-1.35,1.46) | **(1.72,5.34)** | (-1.97,0.51) | (-0.77,1.61) |
| | Disabled (64) | (-1.47,1.35) | - | **(1.64,5.28)** | (-2.19,0.57) | (-1.10,1.79) |
| | Frozen | **(-5.33,-1.69)** | **(-5.33,-1.64)** | - | **(-5.93,-2.58)** | **(-4.64,-1.48)** |
| | Unfrozen | (-0.53,1.93) | (-0.54,2.19) | **(2.58,5.97)** | - | (-0.10,2.52) |
| | Uninitialized | (-1.62,0.78) | (-1.79,1.03) | **(1.50,4.70)** | (-2.53,0.11) | - |

