# OpenReview forum: "Temporally Rich Deep Learning Models for Magnetoencephalography"
_TMLR — Accepted by TMLR_

### Review · Reviewer_uhNr · 2023-09-08

**Summary Of Contributions:**

The paper improves deep learning techniques for Magnetoencephalography (MEG) data analysis. Specifically, it introduces architectures with temporal processing and pre-processing, and evaluates those and previously used architectures on several MEG datasets, including new (for this type of work) ones. The introduced methods improve results on many, although not all, tasks.

**Audience:**

Yes

**Broader Impact Concerns:**

I don't have broader impact concerns with this paper, and don't think it requires a separate section here. However, I'd appreciate if the authors added a paragraph to the discussion/conclusion that outlines the broader implications of this work.

**Claims And Evidence:**

Yes

**Requested Changes:**

**Major**
1. The code should be provided. There are options for anonymous github-like repositories, which could be used during the review period.
2. What is the train/val/test split in 4.1.2 Mother Of Unification Studies? I couldn't find it in the paper, so it should be explained like in the section 4.1.1.
3. On MOUS AudioVis task (Tab. 2), why is there such a big gap between validation and test performance? There's a discussion for LB/UB for that result, but not for val/test.

**Minor**
1. Challenge 3 paragraph: either said twice in "were either exposed to either a visual or auditory stimulus"
2. Page 10 number is broken (somehow includes the link from \cite just at the end of the page?)
3. For all tables: I’d add (ours) after method names so it’s easier to assess the results
4. Appendix tables: the vertical tables are very hard to read. I'd suggest resizing them with \resizebox{\textwidth} to fit on a page horizontally.

**Strengths And Weaknesses:**

I have to note that I don't have experience with MEG data, so my comments might not be very data-specific.

**Strengths**
1. The paper is really well-written: it is easy to read, and it provides detailed explanations for the proposed methods, methods used before, and the specifics of each dataset/task. As a non-expert on MEG data, I found it easy to follow what the authors did and why.
2. Introduction of a temporal component in the architectures is reasonable for this type of data, and this paper tests the idea with several architecturs/datasets. They generally work pretty well, making it a good contribution.
3. The overall evaluation is very extensive, comparing both the proposed networks and several others. Same goes for the dataset/task selection.
4. The discussion of the results is extensive as well, and touches on the data specifics instead of concentrating on pure numbers.

**Weaknesses**
1. No clear one-fits-all architecture: I'd expect default models like ResNet18 to uniformly underperform on the MEG tasks, so the current results (when the proposed models sometimes fall behind) are a bit surprising.
2. No code provided with the submission.

---

### Review · Reviewer_Btfa · 2023-10-17

**Summary Of Contributions:**

The paper offers new DL models for MEG data which capture temporal structure. This extends in a logical and useful way beyond the previous use of CNNs which capture only spatial structure.

The paper points out that spatial structure may not even exist in MEG data, while temporal structure certainly does.

The accuracies of the offered models do not appear to be substantially better than accuracy of prior models. However, they represent a fundamentally stronger basis from which to develop architectures.

The paper includes a strong grounding in the literature of this subfield.

**Audience:**

Yes

**Claims And Evidence:**

Yes

**Requested Changes:**

To the editors of TMLR:
Please make templates include line numbers! They make it so much easier to comment on papers.

MIscellaneous comments: Note: only the item about lack of uncertainty estimates in the main tables is serious. The rest of the items highlight points where an average reader was confused, so extra clarification might help).

Abstract line 3: add an "is"

pg 2: "some of this work" -> "some of this prior work" (for clarity - "this work" can be the present paper)

pg 2: wonderful text

pg 3 Contributions: the 4th item might better go second to match Challenge ordering

pg 4 Background: "All brain activity is the result of electrical currents". Is this true? I thought chemical neurotransmitters and local potential fields (for example) have large functional impacts. Neural spiking (or oscillating) is just one aspect of total brain activity, which is however easily measured and therefore inordinately focused upon.

pg 6 Bottom, pg 7 top: why describe a 2D convolution which is 1 x 1? This is confusing terminology.

pg 7: "1-dimensional kernel in a 2-dimensional convolution": what does this mean? This looks to be just trivially a "convolution", in the sense that 2 * 3 is an n-dimensional "convolution" that multiplies 3 by a kernel of dimension 1 x 1 x 1 x ...

pg8 "pointwise (1 x 1) convolution": ditto - why this terminology? There are many examples of this in the paper.

fig 3: the step in red (with size "t/4") is not in the key in the caption.

pg 8 SERes: it learns a spatial embedding. Why is spatial structure targeted at all? This appears to contradtict the earlier observations that MEG data do not have spatial structure due to how they are collected. Some explanation/motivation would be useful here.

pg 8 3.4: lovely section

pg 9: 3.5 "(temporally enriched)": meaning is unclear.

pg 9 3.5: why is the input compressed by a factor of 2 (t -> t/2)?

4.11 line 4: were these 3 sessions disjoint or continuous (directly following one another)? Also, what was the duration of the pattern/auditory tone?

pg 10 4.11 2nd paragraph: "epoch" is introduced without definition (perhaps add a note, or move up the content below).

pg 11 top (also on pg 10): I'm baffled why the epochs are so short (500 ms - less than one word?). This comes up in the discussion of results (pg 17 near bottom). Why not take a longer epoch to capture multiple words? Please describe the logic that drove this decision.

pg 12 4.2.1: does the use of 3 identical channels effectively increase the number of filters by a factor of 3? If so, what is the implication of increasing the number of parameters in the model?

pg 13 5.1 "statistically significant": this is not assessible given how Tables 2 - 6 are constructed. They need uncertainty estimates. Please see comments in "weaknesses" above.

Tables 7+: These require more explanation to be interpretable. See comments in "weaknesses" above.

**Strengths And Weaknesses:**

Strengths:

The paper is of very high quality. In particular:

The paper is well-written, and the language is precise.

The treatment is thorough, covering a wide array of nuanced detail.

The coverage of prior work, and placing of this paper in context, is excellent (to the degree that I can infer based on my limited knowledge of the domain).

The proposed models are well-motivated. The addition of temporal structure is a fundamentally valuable advance.

The combination of the above enable this to be a potentially foundational paper for this literature.

---------

Weaknesses:

The accuracies of the proposed models do not appear to differ greatly from that of prior models. While this is not of primary importance, it still deserves some careful discussion.

The results tables are relatively uninformative, because they do not include +/- intervals (std dev, 95% CIs, or similar). This makes the relative accuracies hard to interpret - is there really any differerence beyond training noise?

Also, to my mind the bootstrap CI tables do not add value as-is: (a) they are too far from the main results tables, (b) their derivation is not explained, (c) I do not see how to align the values in the two sets of tables, and (d) if they could be aligned it would still require flipping back and forth while holding table values in memory.

In sum, the main results tables should include uncertainty estimates.

---

> ### Comment · Reviewer_Btfa · 2023-11-19
> **I don't see author responses**
>
> Perhaps I'm missing something. I do not see any author responses to two of the reviewers (including me), and the one main sticking point I noted (CIs in the results tables and explanation of the bootstrapping tables/merging of these with the results tables) has not been addressed.
>
> Thanks much.

---

> > ### Author Response · Authors · 2023-11-19
> > **Update visibility**
> >
> > Sorry for the inconvenience and thank you for bringing this to our attention. The default visibility for our response did not include the reviewers for some reason. You should be able to see our response now and we look forward to hearing your comments regarding the confidence intervals.
> >
> > Thank you again.

---

> > > ### Comment · Reviewer_Btfa · 2023-11-22
> > > **Response to Authors' comments**
> > >
> > > Thanks for the responses to all reviewers.
> > >
> > > First, I'm happy to vote to publish this paper. It's very well done in all aspects.
> > >
> > > Second, some optional comments and things the authors might do:
> > > 1. I am concerned that a reader like me may get lost in the uncertainty analysis. This would weaken the paper's impact. Some ideas:
> > >
> > > a) In Appendix A, add an explanation of the tables for clarity and ease (maybe just copy the text from the paper body).
> > >
> > > b) Include +/- std devs in the results tables, even if you have other analyses - it is clear and straightforward.
> > >
> > > c) I did not understand the Win/Loss column and explanation.
> > >
> > > 2. The bibliography is hard to search, because the names used as keys in the text are hard to see in the bib. Options: Use numbers; use only initials for first names; put initials after the surnames.
> > >
> > > 3. To me, the similarity of accuracy results with prior architectures is a non-issue, because the proposed architectures are a stronger foundation for future work.

---

> > > > ### Author Response · Authors · 2023-11-30
> > > > **Further response**
> > > >
> > > > We really appreciate your feedback, thank you.
> > > >
> > > > We are more than happy to include an explanation of the tables in Appendix A and you will see that in the next version of the paper.
> > > > We also agree with you regarding the addition of standard deviations to the results and are more than happy to include them for the subject level accuracy results. In the next revision of the paper, we will add a new column in addition to the lower bound and upper bound columns, but we also have the option to replace the upper bound column with this new column if you think that is a better approach.
> > > >
> > > > For the test accuracy however, displaying a form of error may lead to confusion about what those numbers actually are since the statistical tests that we apply for test accuracy compares the difference in test accuracy between two models.
> > > > Including standard deviations for the test accuracy might lead a reader to reasonably assume that there was a statistical test that applied to a single model.
> > > >
> > > > Our Win/Loss column was intended to aggregate the information of these pairwise statistical tests.
> > > > Since we are looking at the difference in accuracy between two models (A and B), there are three different possible outcomes; either model A can be better than model B, model B can be better than model A or we can not find a difference between the two models.
> > > > We say that in the first case, model A wins and model B loses, and in the second, model A loses and model B wins.
> > > > Our Win/Loss column was simply the total of these outcomes for each model.
> > > > Needless to say, we will revise the explanation of the Win/Loss column in the next revision.
> > > > Alternatively, we can replace the Win/Loss column with the new column we mentioned earlier if that addresses your concerns more concisely.
> > > >
> > > > Lastly, we share your opinion about bibliography style but we are under the impression that this style is required by TMLR.
> > > >
> > > > We will provide an updated version of the paper over the weekend. Thank you again for your suggestions.

---

> > > > > ### Comment · Reviewer_Btfa · 2023-11-30
> > > > > **note re bibliography**
> > > > >
> > > > > 1. If wished, you can edit the .bib file to replace names with initials. Then the same .sty format will give a more readable entry in the bibliography, eg "J.W. Jones etc" instead of "Jessica Wilson Jones".
> > > > >
> > > > > 2. Thanks for your clarifications, and for a great paper!

---

> > > > > > ### Author Response · Authors · 2023-12-16
> > > > > > **update**
> > > > > >
> > > > > > Thank you again for your feedback. After careful consideration, we have decided to remove the Win/Loss column in the most recent version of our paper. We believe that focusing on reporting the statistics for subject accuracy on the test set will provide a clearer insight into our findings while still addressing your concerns.

---

### Review · Reviewer_DMPd · 2023-11-15

**Summary Of Contributions:**

This work proposed three new frameworks, SERes, TimeConv, SETra, to MEG classification. It was evaluated on two benchmark datasets: Cam-CAN, and MOUS.

**Audience:**

Yes

**Claims And Evidence:**

Yes

**Requested Changes:**

1. Encrease the font of figure text, it's hard to read now.

2. May consider to condense the description of challenges and methods, which are well-known. Focus more on the important part, or reduce the number of pages.

**Strengths And Weaknesses:**

Strength:
- I'm happy to see studies on MEG. MEG  is an understudied area that might be important for neurological signal analysis and BCI applications.
- This work introduced the method, and the datasets in detail, which should be encouraged here.

- I appreciate the released code although didn't check it in detail

Weakness:
- Overall, the technical part like timeencoder are not novel, which are a simple variation of CNN.

- The performance is not as good as expected (see Tables 2-3), especially on MOUS dataset.

-It's unclear what's the unique characteristics of MEG data (how it's different from EEG, fNIRS, etc.), so there's no dedicated design in the model for specific MEG processing.

---

> ### Author Response · Authors · 2023-11-23
> **Response to review**
>
> Thank you for your review of our paper.
>
> We appreciate your suggestion to increase the font of the figures and have increased the size of the figures in the revised manuscript.
> In response to your comments about condensing the challenges and methods section, we aimed to balance accessibility for both ML and neuroscience communities.
> This approach led us to include a bit more explanatory content than usual for an ML-centric paper.
> We appreciate the suggestion to include more information for an ML readership on the differences between MEG and other neuroimaging technologies.
>
> We recognize, as noted in our responses to other reviewers, that the performance margins between the evaluated models are not always large.
> However, we are confident in our results and have supported our claims with statistical significance testing.
> For example, in Table 2, both of our statistical tests show that our models outperform the other models evaluated on the Cam-CAN dataset.
> On the other dataset our models do not outperform the existing models, but we would like to point out the difference in the clinical setup of that dataset, where each subject was only exposed to one stimulus modality. This may have influenced the kind of relationships that the models were able to learn despite being conceptually same classification task.
>
> Regarding the novelty of the TimeAutoencoder, we acknowledge your perspective.
> However, we wish to emphasize that the primary role of the TimeAutoencoder is to demonstrate the effectiveness of the temporal aspects of MEG data rather than architectural innovation of that model itself.
>
> Regarding our other technical contribution, we'd note that we also have models inspired by Transformers (SETra) as well as CNNs.
> Additionally, we would like to highlight that our primary contribution lies in applying models specifically designed to capture temporal information to MEG data, with the aim of verifying whether this is in fact useful; the overall results, including MOUS, indicate that the answer is yes, but not straightforwardly so.
>
> We hope that TMLR would be a good venue for this work, given that its mission statement is to focus on "technical correctness over subjective significance, to ensure that we facilitate scientific discourse on topics".

---

### Decision · Action_Editor_rBmo · 2024-01-02

**Recommendation:** Accept as is

**Comment:**

All reviewers appreciate the contributions of the paper and recommend acceptance.

With regards to the ICLR journal to conference track, the recommendations are mixed. Looking through the paper, I would say that the scoping is really well-aligned with TMLR so I would recommend keeping the paper as as TMLR only paper.

**Audience:**

Yes.

**Claims And Evidence:**

Yes.